# Demographic causes of adult sex ratio variation and their consequences for parental cooperation

Luke J. Eberhart-Phillips[1,2], Clemens Küpper[2], María Cristina Carmona-Isunza[3], Orsolya Vincze[4,5], Sama Zefania[6], Medardo Cruz-López[7], András Kosztolányi[8], Tom E.X. Miller[9], Zoltán Barta[5], Innes C. Cuthill[10], Terry Burke [11], Tamás Székely [3,5], Joseph I. Hoffman[1] & Oliver Krüger[1]

The adult sex ratio (ASR) is a fundamental concept in population biology, sexual selection, and social evolution. However, it remains unclear which demographic processes generate ASR variation and how biases in ASR in turn affect social behaviour. Here, we evaluate the demographic mechanisms shaping ASR and their potential consequences for parental cooperation using detailed survival, fecundity, and behavioural data on 6119 individuals from six wild shorebird populations exhibiting flexible parental strategies. We show that these closely related populations express strikingly different ASRs, despite having similar ecologies and life histories, and that ASR variation is largely driven by sex differences in the apparent survival of juveniles. Furthermore, families in populations with biased ASRs were predominantly tended by a single parent, suggesting that parental cooperation breaks down with unbalanced sex ratios. Taken together, our results indicate that sex biases emerging during early life have profound consequences for social behaviour.

[1] Department of Animal Behaviour, Bielefeld University, Morgenbreede 45, 33615, Bielefeld, Germany. [2] Research Group Behavioural Genetics and Evolutionary Ecology, Max Planck Institute for Ornithology, Eberhard-Gwinner-Str. 5, 82319, Seewiesen, Germany. [3] Milner Centre for Evolution, Department of Biology and Biochemistry, University of Bath, Claverton Down, Bath BA2 7AY, UK. [4] Hungarian Department of Biology and Ecology, Babeș-Bolyai University, RO-400006, Cluj Napoca, Romania. [5] MTA-DE Behavioural Ecology Research Group, Department of Evolutionary Zoology, University of Debrecen, Debrecen 4032, Hungary. [6] Department of Animal Biology, Faculty of Sciences, University of Toliara, PO Box 185, Toliara, Madagascar. [7] Posgrado de Ciencias del Mar y Limnología, Unidad Académica Mazatlán, Universidad Nacional Autónoma de México, Ciudad Universitaria, 04510, México D.F., Mexico. [8] Department of Ecology, University of Veterinary Medicine Budapest, Budapest 1078, Hungary. [9] Department of BioSciences, Program in Ecology and Evolutionary Biology, Rice University, MS-170, Houston, TX 77005, USA. [10] School of Biological Sciences, University of Bristol, Bristol BS8 1TQ, UK. [11] Department of Animal and Plant Sciences, University of Sheffield, Sheffield S10 2TN, UK. These authors jointly supervised this work: Tamás Székely, Joseph I. Hoffman, Oliver Krüger. Correspondence and requests for materials should be addressed to L.J.E.-P. (email: luke.eberhart@orn.mpg.de)

Sex ratio variation is a fundamental component of life-history evolution. At conception, birth, and adulthood, the ratios of males to females have long been hypothesised by evolutionary biologists and human demographers as catalysts for social behaviour and population dynamics[1,2]. In particular, the adult sex ratio (ASR) exhibits remarkable variation throughout nature, with birds and mammals tending to have male-biased and female-biased ASRs, respectively[3]. Recent studies also show extreme shifts in ASR, due to climate change, in fish[4], amphibians[5], and dioecious plants[6]. By influencing mate availability, ASR bias can alter social behaviour with divorce, infidelity, and parental antagonism being more frequent in sex-biased populations[7,8]. Moreover, in human societies, ASR variation is linked to economic decisions, community violence, and disease prevalence[9–11]. Yet despite the widespread occurrence of ASR bias and its significance in evolutionary ecology and social science, the demographic source(s) of ASR variation and their ramifications for social behaviour remain unclear[12].

Sex ratio theory is concerned with the adaptive consequences of sex-biased parental allocation to offspring[13,14], with the processes generating sex ratio bias after birth receiving less theoretical and empirical attention[15]. Here, we use a demographic pathway model to quantify ASR variation among avian populations and to determine whether this variation is predominantly caused by sex biases at birth, during juvenile development, or in adulthood. We parameterised our model with detailed individual-based life history data from *Charadrius* plovers—small ground-nesting shorebirds that occur worldwide. Plovers exhibit remarkable diversity and plasticity in breeding behaviour with sex roles during courtship, mating, and parental care varying appreciably among populations both between and within species[16,17]. This behavioural variation, coupled with their tractability in the field (Supplementary Movie 1), allowed us to explore the sources and significance of demographic sex biases among closely related wild populations in the light of social evolution.

Our study reports striking variation in ASR across six plover populations that exhibit similar life histories and ecological traits. Sex differences in the apparent survival of juveniles were the main drivers of ASR bias, with deviations in hatching sex ratio and sex-specific adult survival having negligible contributions. Furthermore, families were predominantly tended by a single parent in populations with biased ASRs, suggesting that parental cooperation breaks down under an unbalanced sex ratio. These results highlight the knock-on effects that early life sex biases can have on population dynamics and social behaviour.

## Results

**Field observations and vital rate estimation**. Over a total of 43 observational years of fieldwork, we monitored the survival, fecundity, and breeding behaviour of 6119 individually marked plovers from six populations of five closely related species worldwide (Fig. 1a). We then employed two-sex stage-structured population matrix models to derive the estimates of ASR at equilibrium from stage- and sex-specific demographic rates of annual survival and reproduction (Fig. 1b)[18]. For each population, the numbers of male and female progeny in our model depended on modal clutch size and hatching sex ratio derived from our field data. Mark-recapture methods were used to estimate the apparent survival of juveniles and adults while accounting for sex differences in detection probability (the term 'apparent survival' indicates that mortality cannot be disentangled from permanent emigration)[19]. Fecundity was derived from a mating function that depended on the extent of polygamy observed in each population and the frequency of available mates (see Methods for details).

**Demographic origin of sex biases**. The hatching sex ratio, based on 1139 hatchlings from 503 families, did not deviate significantly from parity in any of the populations (Fig. 2a). Conversely, sex biases in apparent survival varied considerably within and among species and, in most populations, juvenile survival was more biased than adult survival—either towards males or females (Fig. 2a). Taken together, these sources of demographic sex bias rendered notable deviations in ASR from parity for three populations (two male biased and one female biased; Fig. 2b).

Matrix models provide a flexible analytical environment to decompose the feedbacks between state-dependent vital rates and population response—an important method used in conservation biology for understanding life-history contributions to population growth and viability[20]. In our case, we modified this approach to assess the relative contributions of sex allocation and sex-specific survival on ASR bias[18]. We found that sex biases in apparent survival during the juvenile stage contributed the most to sex ratio bias of the adult population: sex biases in juvenile apparent survival contributed on average 7.8 times more than sex biases in adult apparent survival and 326.6 times more than sex biases at hatching (Supplementary Fig. 1). Moreover, variation in hatching sex ratio had no effect on ASR and remained unbiased even in populations with strong sex differences in juvenile survival. This provides empirical support for Fisher's[13] prediction of unbiased sex allocation regardless of sex-biased survival of independent young or adults. However, we cannot dismiss biased sex allocation at the individual level, which would average out at the population level[21]. This critical test warrants further long-term study.

**Implications for parental cooperation**. In species where both parents have equal caring capabilities, the desertion of either parent is often influenced by the availability of potential mates[22]—parental care by the abundant sex is expected to be greater than that of the scarcer sex due to limited future reproductive potential[23]. Detailed behavioural observations of 471 plover families revealed high rates of parental desertion in populations with biased ASRs, whereas desertion was rare in unbiased populations (Fig. 3). We evaluated our a priori prediction of a quadratic relationship between parental cooperation and ASR variation using a regression analysis incorporating a bootstrap procedure that acknowledged uncertainty in our estimates of ASR and parental care (see Methods for details). We found that families in male- or female-biased populations tended to express higher rates of parental desertion, while unbiased populations were more likely to exhibit parental cooperation (Fig. 3a). This is supported by experimental evidence of sex-biased mating opportunities in three of the populations studied here (Supplementary Fig. 2; ref. [24]). Moreover, the relationship between parental cooperation and local ASR bias was apparent in our within-species contrast of *Charadrius alexandrinus*: the unbiased Cape Verde population exhibited a higher rate of parental cooperation than the male-biased population in Turkey (Fig. 3a). Counterintuitively, we also found a high rate of male-only care in *C. pecuarius* despite ASR being female biased (Fig. 3b), although in line with expectations, *C. pecuarius* also showed the highest proportion of female-only care among our studied populations (Fig. 3b, Supplementary Fig. 3b). This provides partial support for the notion that breeding strategies may respond flexibly to local mating opportunities provided by ASR bias, while also suggesting that other factors may play a role, such as the energetic costs of egg production imposed on females or because of sex differences in parental quality[25] or the age at maturation[26].

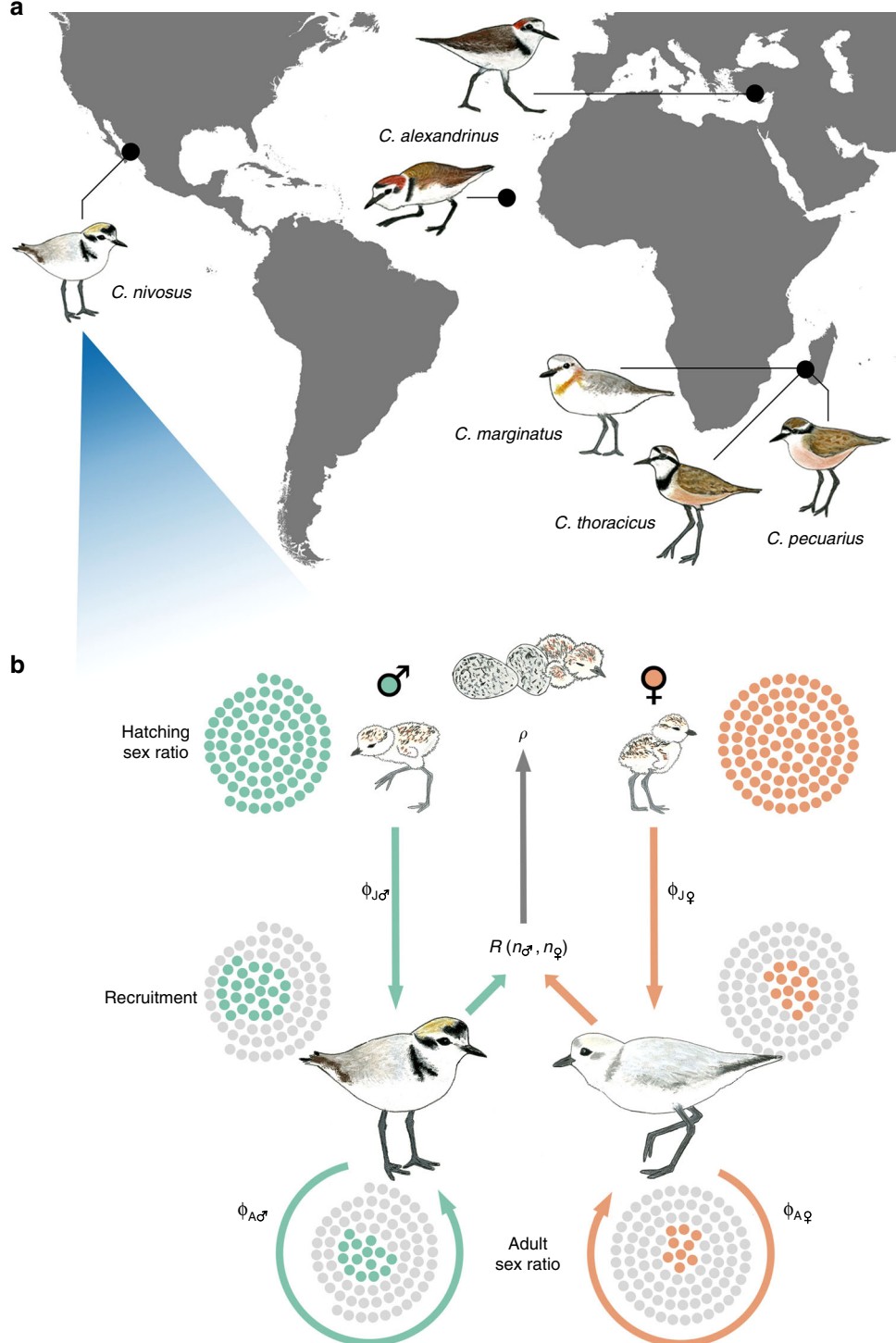

**Fig. 1** Modelling the demographic pathway of adult sex ratio bias in *Charadrius* plovers. **a** Location of the six study populations. *C. pecuarius*, *C. marginatus*, and *C. thoracicus* breed sympatrically in south-western Madagascar, whereas the two populations of *C. alexandrinus* are geographically disparate, inhabiting southern Turkey and the Cape Verde archipelago. The studied *C. nivosus* population is located on the Pacific coast of Mexico. All populations inhabit saltmarsh or seashore habitats characterised by open and flat substrates. **b** Schematic of the stage- and sex-specific demographic transitions of individuals from hatching until adulthood and their contributions to the adult sex ratio (depicted here is *C. nivosus*). The hatching sex ratio ($\rho$, proportion of male hatchlings) serves as a proxy for the primary sex ratio and allocates progeny to the male or female juvenile stage. During the juvenile ('J') stage, a subset of this progeny will survive ($\phi$) to recruit and remain as adults ('A'). Dotted clusters illustrate how a cohort is shaped through these sex-specific demographic transitions to derive the adult sex ratio (mortality indicated by grey dots). The reproduction function, $R(n_{\male}, n_{\female})$, is dependent on mating system and the frequency of available mates (see Methods for details). Original plover illustrations by L.J.E.-P. World map produced from Open Database Licensed shapefiles provided by OpenStreetMapData

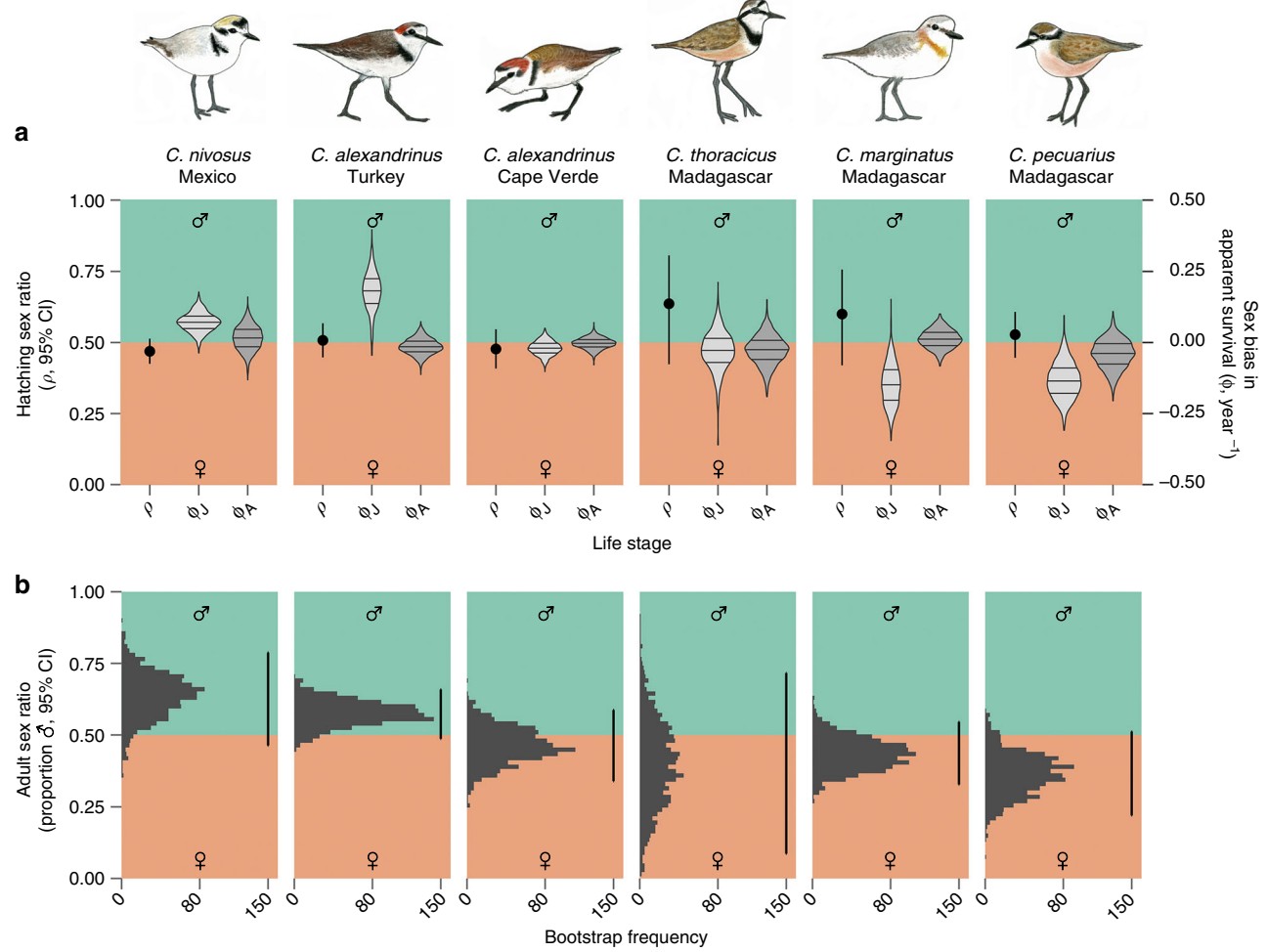

**Fig. 2** Inter- and intra-specific variation in sex-biased demographic rates. **a** Hatching sex ratios of successful clutches (proportion of chicks that are male) are shown as point estimates ($\rho \pm 95\%$ CI; left y-axis), and sex bias (i.e. difference between males and females) in annual apparent survival rates of juveniles ($\phi_J$) and adults ($\phi_A$) is shown as violin plots (right y-axis). Horizontal lines within violin plots indicate the median and interquartile ranges of the bootstrapped estimates (see Methods for details). **b** Bootstrap distributions of the derived ASRs based on the sex- and stage-specific apparent survival rates shown in panel **a**. Vertical bars on the right side of histograms indicate the 95% CI of ASRs based on 1000 iterations of the bootstrap (mean ASR [95% CI]: *C. nivosus* = 0.638 [0.464, 0.788], *C. alexandrinus* [Turkey] = 0.576 [0.487, 0.659], *C. alexandrinus* [Cape Verde] = 0.463 [0.339, 0.587], *C. thoracicus* = 0.401 [0.086, 0.716], *C. marginatus* = 0.434 [0.328, 0.546], *C. pecuarius* = 0.363 [0.220, 0.512]). Original plover illustrations by L.J.E.-P

## Discussion

There are several ways in which sex-biased juvenile survival could arise. Natal dispersal rates may differ between the sexes, as is typical of many birds[27], which could contribute to our estimates of sex-biased apparent survival. Genetic studies of several of the populations presented here are partially consistent with this hypothesis, as island populations of *C. alexandrinus* and the endemic *C. thoracicus* have reduced gene flow relative to comparable mainland populations[28,29]. However, sex-biased juvenile survival in plovers has been reported elsewhere, even after accounting for dispersal[30,31], implying that sex differences in mortality are at least partly driven by intrinsic factors, conceivably for example via genotype–sex interactions[32,33]. An alternative but not necessarily mutually exclusive explanation is that males and females may differ in their premature investment into reproductive traits, which could inflict survival costs for the larger or more ornamented sex[34]. Although sexual dimorphism among adults is negligible in the species we studied[35,36], sex-specific ontogeny does appear to vary in three of these populations. In male-biased populations of *C. nivosus* and *C. alexandrinus*, female hatchlings are smaller and grow more slowly than

their brothers during the first weeks of life, whereas juveniles of the unbiased *C. alexandrinus* population exhibit no such sex-specific differences during early development[35].

The association between sex-specific demography and breeding system evolution represents a causality dilemma because of the feedback that parental strategies impose on ASR bias and vice versa[37]. On the one hand, mating competition and parental care may entail costs via sexual selection that could drive differential survival of males over females and have knock-on effects on ASR[38]. On the other hand, sex-specific survival creates unequal mating opportunities via ASR that may influence mating patterns and parenting strategies[38]. Our study provides empirical support for the latter—sex biases emerge prior to sexual maturity, suggesting that this evolutionary feedback loop is catalysed by intrinsic early-life demographic variation. Moreover, our results add to the growing evidence of unbiased birth sex ratios in nature[15] and provide a comprehensive population-level test of Fisher's[13] original prediction that influential sex biases arise in life-history stages beyond parental control. By unravelling the demographic foundations of ASR bias and their consequences for parental cooperation, we hope to stimulate future studies to

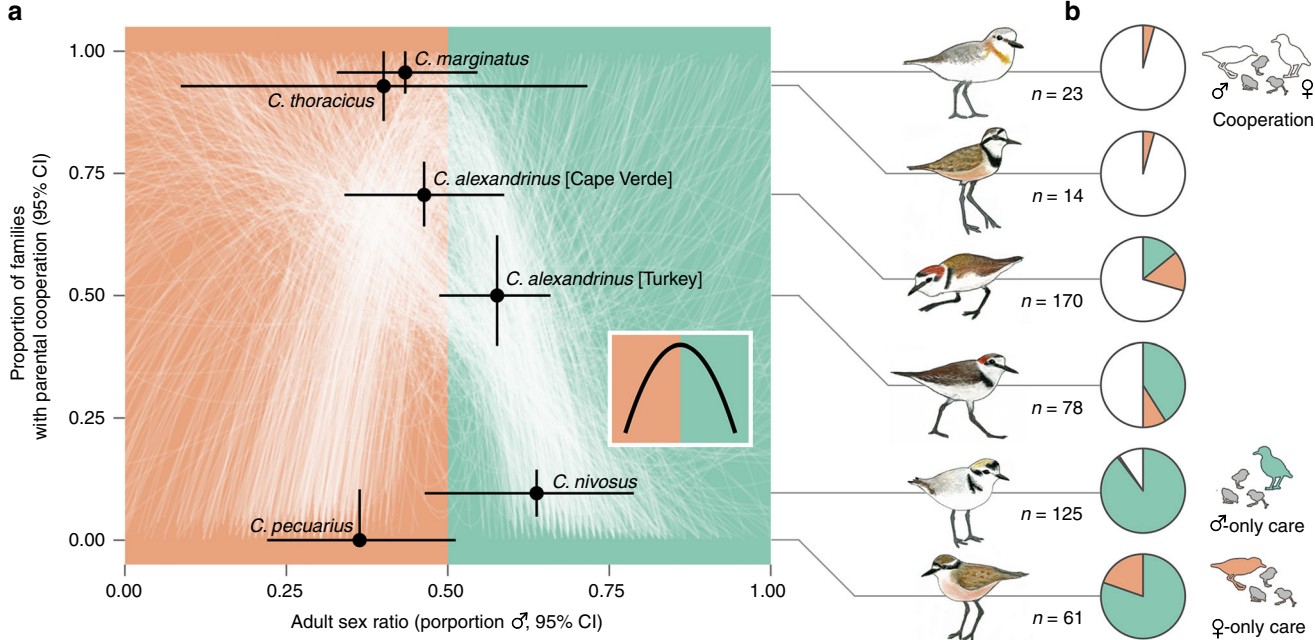

**Fig. 3** Relationship between parental cooperation and the adult sex ratio. **a** Faint white lines illustrate each iteration of the bootstrap, which randomly sampled an adult sex ratio and parental care estimate from each population's uncertainty distribution and fitted them to the a priori quadratic model (shown in inset, Eq. 10). **b** Proportion of monitored plover families that exhibit parental cooperation (white) or single-parent care by males (green) or females (orange). Sample sizes reflect number of families monitored per population. Original plover illustrations and silhouettes by L.J.E.-P

understand the complex relationship between evolutionary demography and behavioural ecology.

## Methods

**Field and laboratory methods.** We studied five *Charadrius* species comprising six populations from four sites worldwide (Fig. 1a, Supplementary Table 1). In Mexico, we monitored the snowy plover (*C. nivosus*) at Bahía de Ceuta, a subtropical lagoon on the Pacific coast. In Madagascar, we monitored the Kittlitz's plover (*C. pecuarius*), white-fronted plover (*C. marginatus*), and the endemic Madagascar plover (*C. thoracicus*), all of which breed sympatrically at a saltwater marsh near the fishing village of Andavadoaka. Lastly, we monitored the Kentish plover (*C. alexandrinus*) at two independent populations located at Lake Tuzla in southern Turkey and at Maio in Cape Verde. The Mexico and Madagascar populations were monitored over a 7-year period, whereas the Turkey and Cape Verde populations were monitored over 6 and 9 years, respectively, thus totalling 43 years of data collection (Supplementary Table 1). Fieldwork was permitted and ethically approved by federal authorities in Cape Verde (Direcção Geral do Ambiente, DGA), Mexico (Secretaría de Medio Ambiente y Recursos Naturales, SEMAR-NAT), Madagascar (Ministry of Environment, Forests, and Tourism of the Republic of Madagascar), and Turkey (Turkish Ministry of Environment).

At each location, we collected mark-recapture and individual reproductive success data during daily surveys over the entire breeding season that typically spanned 3 to 4 months after a region's rainy season. Funnel traps were used to capture adults on broods or nests[39] (Supplementary Movie 1). We assigned individuals to a unique colour combination of darvic rings and an alphanumeric metal ring, allowing the use of both captures and non-invasive resightings to estimate survival (Supplementary Movie 1). Broods were monitored every 5 days on average to assess daily survival and identify tending parents. During captures, approximately 25–50 µL of blood was sampled from the metatarsal vein of chicks and the brachial vein of adults for molecular sex typing.

We extracted DNA from blood samples using an ammonium acetate extraction method[40] and determined sexes with two independent fluorescently labelled sex-typing markers Z-002B[41] and Calex-31, a microsatellite marker on the W chromosome[42]. We amplified sex-typing markers using polymerase chain reaction (PCR) on a DNA Engine Tetrad 2 Peltier Thermal Cycler under the following conditions: 95 °C for 15 min, followed by 35 cycles of 94 °C for 30 s, 56 °C for 90 s, 72 °C for 60 s, and a final extension of 60 °C for 30 min. We visualised the PCR products on an ABI 3730 automated DNA analyser and scored sex-specific alleles using GENEMAPPER software version 4.1 (Applied Biosystems, MA, USA).

**Quantifying parental care.** We evaluated sex role variation by summarising for each population the proportion of all families that were attended bi-parentally or uni-parentally by a male or female. We restricted our field observations to include only broods that were at least 20 days old. Young chicks are attended by both

parents in all populations although, as broods get older, male or female parents may desert the family[43]. As chicks typically fledge around 25 days of age, we therefore choose broods of between 20 and 25 days of age to quantify parental care given that at this age many parents already deserted the family but some still attend the young. Furthermore, we restricted these data to include only broods that had at least two sightings after 20 days. Given these criteria, our dataset consisted of 471 unique families distributed throughout the six populations and pooled across all years of study (Supplementary Table 2). To account for surveyor oversight while recording tending parents (e.g. observing only one parent when two were present), we took a conservative approach by assigning a bi-parental status to families that had both uni-parental and bi-parental observations after the 20th day. In summary, desertion was most common in *C. nivosus* and *C. pecuarius*, whereas bi-parental care was most common in *C. alexandrinus* (Cape Verde), *C. thoracicus*, and *C. marginatus* (Fig. 3b, Supplementary Table 2). The *C. alexandrinus* population in Turkey had 50% bi-parental and 50% desertion (Fig. 3b, Supplementary Table 2). *C. pecuarius* had the highest incidence of male desertion (20%; Fig. 3b, Supplementary Table 2). To acknowledge uncertainty in these parental care proportions given variation in sample size, we used a method that estimated simultaneous 95% confidence intervals according to a multinomial distribution[44] (Supplementary Table 2).

**Estimation of sex- and stage-specific survival.** Our structured population model considered sex-specific survival during two key stage classes in life history: juveniles and adults (Fig. 1b). The juvenile stage was defined as the 1-year transition period between hatching and recruitment into the adult population. The adult stage represented a stasis stage in which individuals were annually retained in the population.

We used mark-recapture models to account for sex, stage, and temporal variation in encounter ($p$) and apparent survival ($\phi$) probabilities as they allow for imperfect detection of marked individuals during surveys and the inclusion of individuals with unknown fates[42]. We use the term "apparent survival" as true mortality cannot be disentangled from permanent emigration in this framework[19]. We used Cormack–Jolly–Seber models to estimate juvenile and adult survival, with 1-year encounter intervals. Juvenile and adult survival models were constructed from design matrices that included sex, year, and stage as factors. Since we were primarily interested in stage- and sex-specific variation in survival, all models included a $\phi \sim sex * stage$ component. Our model selection thus evaluated the best structure explaining variation in detection probability by comparing all interactions between sex, year, and stage (e.g. $p \sim sex * year * stage$). We constructed survival models with the R package "RMark"[45] and estimated demographic parameters via maximum likelihood implemented in program MARK[46]. We evaluated whether our data were appropriately dispersed (i.e. c-hat ≤ 3; ref. [19]) by employing the "median c-hat" goodness-of-fit bootstrap simulation in program MARK[46].

**Estimating hatching sex ratios.** To account for potential sex biases arising prior to the juvenile stage (i.e. sex allocation or sex-specific embryo mortality), we tested

whether the hatching sex ratio deviated significantly from parity. Each population was analysed separately using a general linear mixed effect model with a binomial error distribution and a logit link function (R package "lme4"[47]). In this model, the response variable was chick sex, the fixed effect was the intercept, and brood identifier was included as a random factor to control for the non-independence of siblings from the same nest. Because plover chicks are precocial, post-hatch brood mixing can occur. Consequently, our dataset for analysing hatching sex ratio included only complete broods (i.e. with no missing chicks) that were captured at the nest on the same day of hatching (503 unique families with 1139 chicks, Supplementary Table 3). The fixed-effect intercepts of all populations were not significantly different from zero, indicating that hatching sex ratios did not deviate from parity (Fig. 2a).

**Matrix model structure.** We built two-sex post-breeding matrix models for each plover population that incorporated two annual transitions denoting juveniles and adults (Fig. 1b). The projection of the matrix for one annual time step ($t$) is given by:

$$\mathbf{n}_t = \mathbf{M}\mathbf{n}_{t-1},\tag{1}$$

where $\mathbf{n}$ is a $4 \times 1$ vector of the population distributed across the two life stages and two sexes:

$$\mathbf{n} = \begin{bmatrix} \text{♀Juvenile} \\ \text{♀Adult} \\ \text{♂Juvenile} \\ \text{♂Adult} \end{bmatrix}\tag{2}$$

and $\mathbf{M}$ is expressed as a $4 \times 4$ matrix:

$$\mathbf{M} = \begin{bmatrix} 0 & R_{♀}(1-\rho) & 0 & R_{♂}(1-\rho) \\ \phi_{♀J} & \phi_{♀A} & 0 & 0 \\ 0 & R_{♀}\rho & 0 & R_{♂}\rho \\ 0 & 0 & \phi_{♂J} & \phi_{♂A} \end{bmatrix},\tag{3}$$

where transition probabilities ($\phi$) between life stages are the apparent survival rates of female (♀) and male (♂) juveniles (J) and adults (A). The hatching sex ratio ($\rho$) describes the probability of hatchlings being either male ($\rho$) or female ($1-\rho$), and was estimated for each population from our field data (see above). Per capita reproduction of females ($R_{♀}$) and males ($R_{♂}$) is expressed through sex-specific mating functions used to link the sexes and produce progeny for the following time step given the relative frequencies of each sex[20]. We used the harmonic mean mating function which accounts for sex-specific frequency dependence[48]:

$$R_{♀}(n_{♂}, n_{♀}) = \frac{kn_{♂}}{n_{♂} + n_{♀}h^{-1}}, \; R_{♂}(n_{♂}, n_{♀}) = \frac{kn_{♀}}{n_{♂} + n_{♀}h^{-1}},\tag{4}$$

where $k$ is the modal clutch size (3 in *C. nivosus*, *C. alexandrinus*, and *C. marginatus*, and 2 in *C. thoracicus* and *C. pecuarius*), $h$ is an index of the annual number of mates acquired per male (i.e. mating system, see below), and $n_{♀}$ and $n_{♂}$ are the densities of females and males, respectively, in each time step of the model.

**Quantifying the mating system.** Demographic mating functions are traditionally expressed from the perspective of males[48], whereby $h$ is the average harem size (number of female mates per male). Under this definition, $h > 1$ signifies polygyny, $h = 1$ monogamy, and $h < 1$ polyandry[49]. Although both sexes can acquire multiple mates in a single breeding season, within-season polygamy is typically female biased in plovers, meaning that females tend to have multiple male partners within a season. Thus, in accordance with the predominantly polyandrous or monogamous mating systems seen across these six populations, $h$ was derived by first calculating the mean annual number of mates for each female ($\mu_i$):

$$\mu_i = \begin{cases} 1, & \text{if } \frac{m_i}{b_i} \leq 1 \\ \frac{m_i}{b_i}, & \text{if } \frac{m_i}{b_i} > 1 \end{cases},\tag{5}$$

where $b_i$ is the total number of years female $i$ was seen breeding and $m_i$ is the total number of mating partners female $i$ had over $b_i$ years. If female $i$ tended to have only one mating partner within and between seasons (i.e. $\frac{m_i}{b_i} \leq 1$), $\mu_i$ was set to 1 because they were functionally monogamous. Alternatively, if female $i$ was polyandrous within seasons, $\mu_i$ was greater than 1 to account for the additional fecundity of these extra matings. From this, we calculated $h$ as the inverse of the population average of $\mu_i$:

$$h = \left( \frac{1}{n} \sum_{i=1}^{n} \mu_i \right)^{-1},\tag{6}$$

where $n$ is the total number of females in a given population.

Our dataset to estimate $h$ for each population only included females for which we were confident of the identity of their mates, and had observed them in at least

two reproductive attempts. In summary, $h$ varied among populations (Supplementary Fig. 4), with *C. nivosus* ($h = 0.82$), *C. alexandrinus* (Turkey; $h = 0.85$), and *C. pecuarius* ($h = 0.86$) having more polyandrous mating systems and *C. alexandrinus* (Cape Verde; $h = 0.96$), *C. thoracicus* ($h = 1$), and *C. marginatus* ($h = 0.90$) all having more monogamous mating systems.

**Estimation of ASR.** We estimated ASR from the stable stage distribution ($\mathbf{w}$) of the two-sex matrix model:

$$\text{ASR} = \frac{w_{♂A}}{w_{♂A} + w_{♀A}},\tag{7}$$

where $w_{♂A}$ and $w_{♀A}$ provide the proportion of the population composed of adult males and females, respectively, at equilibrium. To evaluate uncertainty in our estimate of ASR due to sampling and process variation in our apparent survival parameters, we implemented a bootstrapping procedure in which each iteration: (i) randomly sampled our mark-recapture data with replacement, (ii) ran the survival analyses described above, (iii) derived stage- and sex-specific estimates of apparent survival based on the model with the lowest $\text{AIC}_C$ (i.e. $\Delta\text{AIC}_C = 0$; Supplementary Fig. 5), (iv) constructed the matrix model (Eq. 3) of these estimates, (v) derived the stable stage distribution through simulation of 1000 time steps, then (vi) derived ASR from the stable stage distribution at equilibrium on the 1000th time step. This approach ensured that parameter correlations within the matrix were retained for each bootstrap and it also accounted for non-linearity in the mating function. We ran 1000 iterations and evaluated the accuracy of our ASR estimate by determining the 95% confidence interval of its bootstrapped distribution. Note that our method estimated ASR as the asymptotic value predicted under the assumption that each population was at equilibrium and thus we could not evaluate inter-annual variation in asymptotic ASR. Nonetheless, our model-derived ASR estimate of the *C. nivosus* population falls within annual count-based ASR estimates of this population[50], providing support that our method is robust. Count-based estimates of ASR from the remaining populations in our study are unfortunately uninformative due to our limited sample of marked individuals with known sex.

Our mark-recapture analysis was based on the encounter histories of 6119 uniquely marked and molecularly sexed individuals (Supplementary Table 4). After implementing the bootstrap procedure, we found that variation in the encounter probabilities of juveniles and adults is best explained by sex, year, and age in *C. nivosus*, *C. alexandrinus* (Turkey), and *C. pecuarius* (Supplementary Fig. 5). Encounter probability was best explained by age and year in *C. alexandrinus* (Cape Verde) and *C. marginatus* (Supplementary Fig. 5). In *C. thoracicus*, encounter probability was best explained by sex and year (Supplementary Fig. 5). Our mark-recapture data were not over-dispersed (Supplementary Table 4).

**Life table response experiment of ASR contributions.** Perturbation analyses provide information about the relative effect that each component of a matrix model has on the population-level response, in our case ASR. To assess how influential sex biases in parameters associated with each of the three life stages were on ASR dynamics, we employed a life table response experiment (LTRE). An LTRE decomposes the difference in response between two or more "treatments" by weighting the difference in parameter values by the parameter's contribution to the response (i.e. its sensitivity), and summing over all parameters[20]. Our LTRE compared the observed scenario ($\mathbf{M}$), to a null scenario with no sex differences ($\mathbf{M}_0$), whereby all male survival rates were set equal to the female rates ($\mathbf{M}_{0♀}$), the hatching sex ratio was unbiased (i.e. $\rho = 0.5$), and mating system was monogamous (i.e. $h = 1$). Thus, our LTRE identifies the drivers of ASR bias by decomposing the absolute parameter contributions to the difference between the ASR predicted by our model and an unbiased ASR[18]. To verify if our method was robust, we also evaluated a null scenario in which all female survival rates were set equal to the male rates ($\mathbf{M}_{0♂}$).

The LTRE contributions ($C$) of a sex bias in stage-specific apparent survival ($\phi_i$), mating system ($h$), and hatching sex ratio ($\rho$) in $\mathbf{M}$ were calculated following Veran and Beissinger[18]:

$$\mathbf{M}_{0♀} \text{ scenario}: C(\phi_i) = \left( \phi_{♂i} - \phi_{♀i} \right) \times \left. \frac{\partial \text{ASR}}{\partial \phi_{♂i}} \right|_{\mathbf{M}'}$$

$$\mathbf{M}_{0♂} \text{ scenario}: C(\phi_i) = \left( \phi_{♀i} - \phi_{♂i} \right) \times \left. \frac{\partial \text{ASR}}{\partial \phi_{♀i}} \right|_{\mathbf{M}'}$$

$$\mathbf{M}_{0♀} \text{ and } \mathbf{M}_{0♂} \text{ scenarios}: C(h) = (h - 1) \times \left. \frac{\partial \text{ASR}}{\partial h} \right|_{\mathbf{M}'}$$

$$\mathbf{M}_{0♀} \text{ and } \mathbf{M}_{0♂} \text{ scenarios}: C(\rho) = (\rho - 0.5) \times \left. \frac{\partial \text{ASR}}{\partial \rho} \right|_{\mathbf{M}'},\tag{8}$$

where $\left. \frac{\partial \text{ASR}}{\partial \theta} \right|_{\mathbf{M}'}$ is the sensitivity of ASR to perturbations in the demographic rate $\theta$ (i.e. $\phi_i$, $h$, or $\rho$) of matrix $\mathbf{M}'$, which is a reference matrix "midway" between the

observed and the null scenarios[18]:

$$\mathbf{M}' = \frac{\mathbf{M} + \mathbf{M}_0}{2}. \tag{9}$$

The two-sex mating function makes our model non-linear in the sense that the projection matrix, and specifically the fecundity elements (Eq. 4), depends on sex-specific population structure. Perturbation analyses must therefore accommodate the indirect effects of parameter perturbations on population response via their effects on population structure, such as the relative abundance of males and females which can affect mating dynamics and fecundity. To estimate the sensitivities of the ASR to vital rate parameters, we employed numerical methods that independently perturbed each parameter of the matrix, simulated the model through 1000 time steps, and calculated ASR at equilibrium. This produced parameter-specific splines from which $\frac{\partial \mathrm{ASR}}{\partial \theta}\big|_{\mathbf{M}'}$ could be derived. This approach appropriately accounts for the non-linear feedbacks between vital rates and population structure, though it does not isolate the contribution of this feedback[49,51]. Under either scenario (i.e. $\mathbf{M}_{0\female}$ or $\mathbf{M}_{0\male}$), our LTRE revealed that across all populations, sex differences in juvenile apparent survival made the largest overall contribution to ASR bias (Supplementary Fig. 1). Likewise, for all populations, sex biases at hatching and in mating system had negligible effects on ASR variation (Supplementary Fig. 1).

**Evaluating the association between ASR bias and parental cooperation**. To test the relationship between ASR bias and parental cooperation, we conducted a regression analysis of the following quadratic model:

$$P_{\male\female} = \beta_0 + \beta_1 A + \beta_2 A^2 + \varepsilon, \tag{10}$$

where $P_{\male\female}$ is the proportion of families exhibiting parental cooperation, $\beta_i$ are the regression parameters (i.e. intercept and coefficients), $A$ is the ASR, and $\varepsilon$ is random error. We chose a quadratic model a priori as we expected maximum parental cooperation at unbiased ASR but minimum cooperation at both male- and female-biased ASRs (see inset in Fig. 3a). This relationship was assessed with a bootstrap procedure that incorporated uncertainty in our estimates of ASR and parental care. Each iteration of the bootstrap (i) randomly sampled an ASR value from the 95% confidence interval of each population shown in Fig. 2b, (ii) randomly sampled a parental care value from the truncated 95% confidence interval of each population shown in Supplementary Table 2, then (iii) fitted the regression model. We ran 1000 iterations of the bootstrap and evaluated overall relationships by visualising the central tendency of the regressions. We also evaluated the relationship between ASR variation and male-only or female-only care using a similar bootstrap procedure of the following models:

$$P_{\male} = \beta_0 + \beta_1^A + \varepsilon, \; P_{\female} = \beta_0 - \beta_1^A + \varepsilon, \tag{11}$$

where $P_{\male}$ and $P_{\female}$ are the proportions of families exhibiting male-only or female-only care, respectively. In this case, we chose exponential models a priori as we expected a non-linear increase in uni-parental care by the abundant sex under biased ASR (Supplementary Fig. 3a). This analysis demonstrated that male-only care tended to be more common in populations with male-biased ASR (mean $\beta_1$ = 0.551 [−0.849, 1.559 95% CI]) and female-only care tended to be more common in female-biased populations (mean $\beta_1$ = −0.149 [−0.427, 0.082 95% CI]; Supplementary Fig. 3b). However, the overall magnitude of the effect of ASR variation on female-only care was less than that of male-only care.

**Code availability**. All of our modelling and statistical analyses were conducted using R version Kite-eating Tree[52] with significance testing evaluated at $\alpha = 0.05$. We provide all computer code and documentation as a RMarkdown file (Supplementary Data 1). This can be downloaded from our GitHub repository: https://github.com/leberhartphillips/Plover_ASR_Matrix_Modeling.

**Data availability**. We provide all the raw datasets needed to reproduce our modelling and analyses. These can be downloaded from our GitHub repository: https://github.com/leberhartphillips/Plover_ASR_Matrix_Modeling.

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

## Acknowledgements

Financial support was provided by a Deutsche Forschungsgemeinschaft (DFG) doctoral fellowship to L.J.E.-P. and supervised by O.K., J.I.H., and T.S. (GZ: KR 2089/9-1). Fieldwork was funded by Sonoran Joint Venture, Tracy Aviary Conservation Fund, Convocatoria de Investigación Científica Básica (CONACyT, Grant no. 157570), Fundação Maio Biodiversidade (FMB), DGA, Camara Municipal do Maio, DFG Mercator Visiting Professorship awarded to T.S., Natural Environment Research Council grant (GR3/10957), and a Biotechnology and Biological Sciences Research Council grant (BBS/B/05788) awarded to I.C.C. and T.S. M.C.C.-I.: CONACyT (PhD scholarship 216052/311485). O.V.: Hungarian Eötvös Scholarship (Tempus Public Foundation MÁEÖ2017_16/156845), Campus Mundi scholarship (CMP-69- 2/2016), and Hungarian Research Fund (OTKA #K113108). A.K.: János Bolyai Research Scholarship of the Hungarian Academy of Sciences and NKFIH grant (K112670, NN125642). T.B.: Leverhulme Fellowship. T.S.: NKFIH-2558-1/2015, ÉLVONAL-KKP 126949, and the Wissenschaftskollegium zu Berlin. Our manuscript benefitted from the constructive comments of S. Beissinger, W. Forstmeier, M. Galipaud, M. Jennions, P. Korsten, A. Shah, and M. Stoffel. We thank N. dos Remedios and K. Maher for molecular work and we are grateful for the generous assistance of countless students, volunteers, and colleagues who aided with fieldwork.

## Author contributions

L.J.E.-P., T.S., J.I.H., and O.K. conceived the study. L.J.E.-P., C.K., M.C.C.-I., O.V., S.Z., A.K., M.C.-L., I.C.C., Z.B., and T.S. planned and collected the field data. L.J.E.-P., C.K., J.I.H., and T.B. performed or supervised the molecular sexing. L.J.E.-P., T.E.X.M., and O.K. implemented the demographic modelling. L.J.E.-P. wrote the manuscript and the RMarkdown file. All authors contributed substantially to revisions of the paper.

## Additional information

**Competing interests:** The authors declare no competing interests.

