## [Peer Review File · Nature Communications]

Reviewers' comments:

Reviewer #1 (Remarks to the Author):

The manuscript summarizes statistical analysis and bootstrapping results of an impressive compilation of avian data with the aim to show variations in adult sex ratio (ASR), variations in parental desertion, and correlations between both.

The presentation of methods and results are is well done and should ensure reproducibility, the results and conclusions drawn are novel and interesting for both scientists within the field and broader readership. Overall, my comments for improving the ms are minor and can be found below:

l 43 and 67: I find the statements that life-history origins of ASR-variation are unclear too strong, because it has been known for decades that survival rates differ between females and males and those differences are an obvious driver of ASR-variation.

p5: sequence of referenced supplemental figures doesn't match their appearance in the main text. Re-order figs to match.

l 122: to support this claim, the ASR should be given in the plots, for example above the panels. I further suggest to either add species names in the plot such that *C. pecuarius* can instantly be located in the plot or describe its position, for example with 'lowest' or '6th in plot'.

l 232 'table S4': you might mean table S3.

l 293: table S3 should be S4. Same in l 299

l 308: Please add info on whether conclusions would change when survival rates would all be set to the female rates.

fig 1 and fig S1 do not match although both are given to support stage-structured modelling. Fig 1 reads as if there are three stages (hatchling, juvenile, adult) and ϕ_{juv} gives the survival from hatching to becoming a juvenile. But ϕ_{juv} in the model and fig S1 actually is the survival from juvenile stage to adult stage. Similarly, in fig 1 ϕ_{ad} is the survival from juvenile to adult stage, but in the model and in fig S1 ϕ_{ad} is the survival of an adult to the next time step.

all figs: the color code orange-green might be hard to decipher for color-blind people, maybe chose a set of colors that is easily to be distinguished by everyone.

fig 2a: the y-label should be hatching sex ratio and survival rates

fig 3: I very much like this figure but I am confused about the bootstrapping trend point. Panels a and b are clearly integrated, but I would expect that the mean (solid black dot) would lie in the half of the plot in 3a which corresponds to the sex that is mostly providing care when the partner deserted. For example, the first and lowest black dot should lie in the green part. Maybe I misinterpret this plot. In any case, an explanation or rectification is needed.

fig S1: The legend says (and in fact the model itself is structured that way) that both sexes are linked via a frequency-dependent mating function R . However, this link is not clear as R is as R_{female} and R_{male} , but in fact, it should be $R_{\text{female,male}}$ or something similar. At the moment the figure shows that offspring production is not resulting from interactions between the sexes at all. This should be

rectified.

I 548 'survivorships did not differ between the sexes': here the information is needed that survival rates have been set to male values. What would change if set to female values?

Reviewer #2 (Remarks to the Author):

This study uses extensive field data from six species of birds to quantify the contributions of life-history variables to observed adult sex ratios (ASR) and patterns of parental care. Juvenile survival is identified as the ASR's main determinant, and balanced ASRs are found to be associated with biparental cooperation. This is an exciting and timely topic, and the manuscript is clearly written and well presented. I only have a few comments.

My main issue is that I can't make sense of eq. 5. The purpose of 'h' is to parametrize a discrete-time matrix model that projects population size and composition from one year to the next. For this purpose, it should be completely irrelevant if birds keep the same partner for multiple years, or whether they find a new partner each year. Yet, the calculation of h works in such a way that keeping the same partner reduces the value of h.

Also, the perturbation analysis was a bit unclear to me. In line 321, the authors refer to "indirect effects of parameter perturbations on population growth." Here I was wondering: do they really mean population growth, or do they mean ASR? Then they go on to say (line 322): "To estimate the sensitivities of the vital rate parameters to ASR..." Here I was wondering: do they really mean "sensitivities of the ASR with respect to vital rate parameters"? My understanding is that sensitivities are calculated for model outputs with respect to model inputs, rather than the other way round.

Finally, the authors might profitably look at a recent paper (Fromhage & Jennions 2016; Nature Comm 7: 12517) that gives reasons for why the cause of ASR variation may actually matter. On the other hand, Kokko & Jennions' 2008 prediction that "sex-specific survival creates unequal mating opportunities via ASR that may influence mating patterns and parenting strategies" was in part based on a technical flaw in their model. The corrected version by Fromhage & Jennions predicts no such effect for sex-specific survival during time-in, i.e., while searching for mates.

Reviewer #3 (Remarks to the Author):

This is a nice study using long-term data from six species of shorebird (one population per species). The authors show that bias in the adult sex ratio (ASR), when it occurs, is consistently caused by sex differences in juvenile mortality or emigration. There is some evidence that populations/species with a biased sex ratio have higher rates of uniparental care. Surprisingly, however, it is not generally the case that the most common sex is the caring parent.

This latter point suggests that there is no (simple) causal relationship between ASR and parental care. The quadratic model is perhaps predicted a priori. But the conclusion that bias in ASR causes breakdown of cooperation between parents rely on the assumption that there is no alternative explanation for why *A. pecuarius* has obligate single-parental care. But does not an alternative seem much more likely given that the expected result based on ASR (biased towards females) is female-only care, and not male care as is apparently the case? In fact, the stronger a priori expectation that the common sex will be more likely to care is not supported by the data. This makes me doubt whether the logic holds up.

ASR could still be important for parental cooperation, of course. But with five data points, and an unexpected direction with respect to which sex that care for offspring, the conclusion that 'parental cooperation breaks down with unbalanced sex ratio' (Abstract) does not really seem to be supported by data.

Nevertheless, this is an elegant study (with beautiful figures!) and the hypothesis that the relationship between ASR and parental care is causal is interesting (but obviously itself not novel).

Minor comments

Line 66. What do you mean by 'life history origin(s) of ASR variation'? The stage at which mortality occurs?

Line 89. I do not mind the term 'apparent survival' (and it does not seem crucial for the aims to that mortality is the cause of biased ASR), but the authors occasionally slip into using 'survival' (e.g., line 128) or 'survivorship' (e.g., line 547). It may be clumsier to read, but consistency is probably a good thing here.

Line 135. What do you mean by 'genotype-sex interactions'? Whatever it means, it seems entirely speculative (I could think of many reasons why males and females differ in survival or dispersal, none of which have to do with their genotype in any more interesting way than that females are ZW and males ZZ).

Methods. I was expecting to see an analyses of how much the ASR varies across years. It may be that the results are there, but if so they do not seem easy to extract from the text or tables. Annual variation in ASR seems like a very important means to also make more out of the relationship to parental care, although I do realise that the sample size is quite small.

Response to NCOMMS-17-19269-T Decision Notification

Title: "Demographic causes of adult sex ratio variation and their consequences for parental cooperation"

Tracking #: NCOMMS-17-19269-T

Authors: Eberhart-Phillips et al.

Note: our responses are shown in *italics* following each reviewer's comment, and edits are shown in ***bold italics***. Revised versions of figures and their legends are found at the end of this document, starting on page 13.

Reviewer #1 (Remarks to the Author):

1.1) The manuscript summarizes statistical analysis and bootstrapping results of an impressive compilation of avian data with the aim to show variations in adult sex ratio (ASR), variations in parental desertion, and correlations between both. The presentation of methods and results are well done and should ensure reproducibility, the results and conclusions drawn are novel and interesting for both scientists within the field and broader readership.

We appreciate the positive comments from Reviewer 1 regarding the reproducible documentation of our analysis and the far-reaching implications of our study, thank you.

Overall, my comments for improving the ms are minor and can be found below:

1.2) l 43 and 67: I find the statements that life-history origins of ASR-variation are unclear too strong, because it has been known for decades that survival rates differ between females and males and those differences are an obvious driver of ASR-variation.

We agree with Reviewer 1 that sex differences in survival have been previously described and that the idea of survival differences emerging at different life history stages is not particularly new. However, here we provide the best empirical evidence of this phenomenon to date using field data across several closely related taxa. Nonetheless, we have modified our original wording to tone down the use of "life-history origins of ASR variation" throughout the manuscript:

*"...despite the widespread occurrence of ASR bias and its significance in evolutionary ecology and social science, the **demographic source(s)** of ASR variation and their ramifications for social behaviour remain unclear" (lines 63-66)*

*"...unravelling the **demographic foundations** of adult sex ratio bias and their consequences..." (line 161)*

1.3) p5: sequence of referenced supplemental figures doesn't match their appearance in the main text. Re-order figs to match.

We thank Reviewer 1 for noticing this inconsistency. We have reorganized the supplementary figures and legends to match their order of appearance in the text. (pages 31-37 of the revised manuscript)

1.4) l 122: to support this claim, the ASR should be given in the plots, for example above the panels. I further suggest to either add species names in the plot such that *C. pecuarius* can instantly be located in the plot or describe its position, for example with 'lowest' or '6th in plot'.

We have followed Reviewer 1's advice and added the species names to Figure 3a to facilitate easy comparison with Figure 2 (see revised version below and on page 26 of the revised manuscript). Note that Figure 3 was revised in light of comments 1.12 from Reviewer 1 and 3.2 from Reviewer 3, yet the comment here about clearly labelling the species is still relevant.

However, concerning Reviewer 1's first point, the ASR values are easily read off the x-axis of this plot and thus we feel that adding these as text to the plot itself would not improve the clarity of the figure but rather make it more cluttered. As such, we prefer to keep the figure as

is, although we would be willing to include the ASR values at a later date if this is considered important.

1.5) | 232 'table S4': you might mean table S3.

Thanks for pointing this out. We corrected this (line 240).

1.6) | 293: table S3 should be S4. Same in | 299

Thanks for pointing this out. We corrected this (lines 310 and 316).

1.8) | 308: Please add info on whether conclusions would change when survival rates would all be set to the female rates.

This is a reasonable request. To address this, we have added a female-only version of the LTRE to the manuscript. The code for this additional analysis has been amended to our RMarkdown (see pages 47-63 of the RMarkdown document). We have also added the results of the "Female-only" scenario to the original "Male-only" scenario as a separate panel in Supplementary Figure 1 (refer to page 33 of the revised manuscript and see below).

*In summary, setting the M_0 matrix to female-only rates gives almost identical results to the original male-only scenario (the only difference being that male-biased parameters contribute slightly more under the female-only scenario, but that female-biased parameters contribute slightly more under the male-only scenario). However, the overall trend remains robust: sex-biases in juvenile survival contribute most to ASR variation. This is because, as described in our methods, the LTRE decomposes which parameters drive the difference between the ASR predicted by the sex-specific model and the ASR predicted by a null model (i.e., no sex-differences, hence $ASR = 0.5$). It is therefore not important which values are chosen for the M_0 matrix, as long as the female values are the same as the male values. The M' matrix, from which the sensitivities are derived from, is simply a matrix "mid-way" between a null scenario and the sex-specific scenario (see Equation 9 on pg. 15). For more details, please see pp. 133 of Veran and Beissinger (2009, *Ecol. Let.* 12:129-143), from which we developed our methods for the two-sex LTRE.*

In addition to modifying Supplementary Figure 1, we clarified that we tested both male-only and female-only scenarios by adding the following text to the manuscript:

*"We compared the observed scenario (M), to a hypothetical scenario (M_0) whereby all female survival rates were set equal to the male rates (**or vice versa**) and the hatching sex ratio was unbiased (i.e. $\rho = 0.5$)." (lines 324-326)*

*"These results are based on a life-table response experiment (LTRE) **that compared the empirically-derived sex-specific model to hypothetical scenarios with no sex differences in demographic rates (top panel: female-only rates, bottom panel: male-only rates)**". (lines 578-580)*

1.9) fig 1 and fig S1 do not match although both are given to support stage-structured modelling. Fig 1 reads as if there are three stages (hatchling, juvenile, adult) and ϕ_{juv} gives the survival from hatching to becoming a juvenile. But ϕ_{juv} in the model and fig S1 actually is the survival from juvenile stage to adult stage. Similarly, in fig 1 ϕ_{ad} is the survival from juvenile to adult stage, but in the model and in fig S1 ϕ_{ad} is the survival of an adult to the next time step.

This is a useful criticism and we appreciate Reviewer 1 noticing this unintentional point of confusion in our figure. To address this, we have modified Figure 1b to more clearly visualize the matrix model used in our analysis (see the modified version below and on page 24 of the revised manuscript). Reviewer 1 correctly pointed out that there are only two sex-specific life stages in our demographic model: hatchlings (juv) and adults (ad). Hatchlings are produced by adults (R), recruit into the adult stage via ϕ_{juv} , and remain in the adult stage (i.e., stasis) via ϕ_{ad} . We have also edited the figure caption to describe the figure clearly:

“Figure 1. Modelling the demographic pathway of adult sex ratio bias among plovers worldwide. (a) Location of the six study populations. *C. pecuarius*, *C. marginatus*, and *C. thoracicus* breed sympatrically in south-western Madagascar, whereas the two populations of *C. alexandrinus* are geographically disparate, inhabiting southern Turkey and the Cape Verde archipelago. The studied *C. nivosus* population is located on the Pacific coast of Mexico. All populations inhabit saltmarsh or seashore habitats characterized by open and flat substrates. (b) Schematic of the stage- and sex-specific demographic transitions of individuals from hatching until adulthood and their contributions to the adult sex ratio (depicted here is *C. nivosus*). The hatching sex ratio (ρ , **proportion of male hatchlings**) serves as a proxy for the primary sex ratio and allocates progeny to the male or female juvenile stage. During the juvenile (‘juv’) stage, a subset of this progeny will survive (ϕ) to recruit and remain as adults (‘ad’). **Dotted clusters illustrate how a cohort is shaped through these sex-specific demographic transitions to derive the adult sex ratio (mortality indicated by grey dots). The reproduction function, $R(n_{\sigma}, n_{\phi})$, is dependent on mating system and the frequency of available mates (see Methods for details).**” (lines 527-541)

Furthermore, we have decided to remove Supplementary Figure 1 from the original manuscript because the modifications made to Figure 1 have made it redundant (see also comment 1.13 below). We have also added species names to Figure 1a for easier comparison with other figures.

1.10) all figs: the color code orange-green might be hard to decipher for color-blind people, maybe chose a set of colors that is easily to be distinguished by everyone.

We thank Reviewer 1 for voicing this concern. Exactly for this reason, we used a simulator (<http://www.color-blindness.com/coblis-color-blindness-simulator/>) in combination with help from a colour-blind colleague to find an appropriate colour palette to use in our original figures. We therefore believe this is an appropriate palette in comparison to the stereotypical blue-pink scheme often used to represent males and females.

1.11) fig 2a: the y-label should be hatching sex ratio and survival rates

This figure has a double y-axis: on the left side is the title for hatching sex ratio (ρ), and on right side is the title for the sex-bias in survival (ϕ). Since both axes straddle a common center (i.e., parity in hatching sex ratio or no sex-difference in survival) we believe that including both axes in the plot allows for better comparability of the skew in these demographic parameters. To make this more evident, we have added the following text to the figure caption:

“(a) Hatching sex ratios of successful clutches (proportion of chicks that are male) are shown as point estimates ($\rho \pm 95\%$ CI; **left y-axis**), and sex bias (i.e. difference between males and females) in annual apparent survival rates of juveniles (ϕ_{juv}) and adults (ϕ_{ad}) **are shown as violin plots (right y-axis).**” (lines 542-546)

1.12) fig 3: I very much like this figure but I am confused about the bootstrapping trend point. Panels a and b are clearly integrated, but I would expect that the mean (solid black dot) would lie in the half of the plot in 3a which corresponds to the sex that is mostly providing care when the partner deserted. For example, the first and lowest black dot should lie in the green part. Maybe I misinterpret this plot. In any case, an explanation or rectification is needed.

We agree with Reviewer 1 that Figure 3 was confusing at first glance. The main result to take away from the figure is that populations exhibiting an unbiased ASR (i.e., 95% CI encompass an ASR of 0.5) tend to show the highest rates of parental cooperation. There is undoubtedly some unexplained variation in the sex-specific desertion rates, but the overall pattern of $P(\text{cooperation}) \sim \text{ASR-bias}$ remains. The confusion expressed by Reviewer 1 stems from the unexpectedly high rate of female desertion in *C. pecuarius* given its female-biased ASR (i.e., the species on the bottom). This is a similar point raised by Reviewer 3 (see comment 3.2 below). We discuss this peculiarity in the main text on lines 125-128 of the revised manuscript. To clarify this, we have also added an inset to Figure 3a to show our a priori quadratic prediction of the relationship between parental cooperation and ASR bias. We have also

decided to remove the original bold average trend line from Figure 3a because with 6 data points we would like to acknowledge the limited power of this analysis (i.e., we encourage readers to explore the variation in the white bootstrap iterations and come to their own conclusion of this relationship). Furthermore, in our revised version of this figure/analysis, we have calculated 95% CIs for the point estimates of parental cooperation (i.e., the response variable in this analysis) and have subsequently redone the bootstrap procedure to sample across this uncertainty distribution in addition to the uncertainty distribution in ASR. This way we appropriately recognize uncertainty in both the predictor and response variables. See below for the revised version of Figure 3 and page 26 of the revised manuscript.

1.13) fig S1: The legend says (and in fact the model itself is structured that way) that both sexes are linked via a frequency-dependent mating function R. However, this link is not clear as R is as R_female and R_male, but in fact, it should be R_female,male or something similar. At the moment the figure shows that offspring production is not resulting from interactions between the sexes at all. This should be rectified.

We agree with Reviewer 1 that the phrase "...linked via..." is misleading here. R is a function of mating system and the frequency of males and females at a given time step (i.e., $R(n_{\sigma}, n_{\varphi})$). To clarify this we have modified Equation 4 (line 261) to read:

$$R_{\varphi}(n_{\sigma}, n_{\varphi}) = \frac{kn_{\sigma}}{n_{\sigma} + n_{\varphi} h^{-1}}, \quad R_{\sigma}(n_{\sigma}, n_{\varphi}) = \frac{kn_{\varphi}}{n_{\sigma} + n_{\varphi} h^{-1}}$$

Also, we have also edited the description of R in the legend of Figure 1:

"The reproduction function, $R(n_{\sigma}, n_{\varphi})$, is dependent on mating system and the frequency of available mates (see Methods for details)." (lines 540-541)

Additionally, as mentioned above in comment 1.9, we have removed the original Supplementary Figure 1.

1.14) | 548 'survivorships did not differ between the sexes': here the information is needed that survival rates have been set to male values.

This is a good point. As mentioned in point 1.8 above, we have added an additional analysis that tested the LTRE with female-only rates. We added the results of this additional analysis to the revised Supplementary Figure 1 (see below and on page 33 of the revised manuscript) and reworded the figure legend to read:

"These results are based on a life-table response experiment (LTRE) which compared the empirically-derived sex-specific model to hypothetical scenarios with no sex-differences in demographic rates (top panel: female-only rates, bottom panel: male-only rates)." (lines 578-580)

1.15) What would change if set to female values?

We tested this. Please see our response to comment 1.8 above for a detailed explanation.

Reviewer #2 (Remarks to the Author):

2.1) This study uses extensive field data from six species of birds to quantify the contributions of life-history variables to observed adult sex ratios (ASR) and patterns of parental care. Juvenile survival is identified as the ASR's main determinant, and balanced ASRs are found to be associated with biparental cooperation. This is an exciting and timely topic, and the manuscript is clearly written and well presented. I only have a few comments.

We thank Reviewer 1 for their positive response. However, as described in the "Results and Discussion" section (lines 80-82), our study contains six populations comprising five species (i.e., two populations were of the same species, *C. alexandrinus*).

2.2) My main issue is that I can't make sense of eq. 5. The purpose of 'h' is to parametrize a discrete-time matrix model that projects population size and composition from one year to the next. For this purpose, it should be completely irrelevant if birds keep the same partner for multiple years, or whether they find a new partner each year. Yet, the calculation of h works in such a way that keeping the same partner reduces the value of h.

In fact, our model operates in exactly the way that Reviewer 2 intuitively it should: it does not matter whether a female with one mate chooses the same or a different mate across years. In either case, the mating system is functionally monogamous (i.e., $h=1$) in the sense that every individual has one opposite-sex mate in each breeding season, regardless of the mate's identity. We realize that our original text was not clear on this and have modified the description about this important technical detail in lines 266-286 of our revised manuscript.

Our methods here are necessarily a bit complicated because classic theory for two-sex population models assumes one mating event per time step (i.e., per year in our model) and therefore does not accommodate the rich natural history of our study system, which includes multiple within-year nesting events. We needed to develop a new approach to translate our within-year data on mating system variation into meaningful estimates for mating system on annual time steps. We suspect that the confusion arises from the discrimination of between-season mating system versus within-season mating system, and we have resolved this confusion in our revision.

We start by quantifying the within-season mating system, which is relevant for our population model because it determines how many progeny are produced in a given annual time step (e.g., polyandrous females with two mates within a single season produce 2x more progeny than monogamous females with one mate within a single season). Equation 5 of the revised manuscript (line 274) returns the average number of unique mates per year per female (μ), and hence this is a measure of the within-season mating system.

To help piece this apart, there are three scenarios to consider:

- 1) *For an individual with the same partner over multiple years (i.e., monogamous within and between seasons), μ will be <1 .*
- 2) *For an individual with one new partner each year (i.e., monogamous within seasons but polygamous between seasons), μ will be $=1$.*
- 3) *For an individual with more than one partner each year (i.e., polygamous both within and between seasons), μ will be >1 .*

Please refer to Supplementary Figure 4 to see the raw data of individuals in these scenarios among populations. Our next step was to derive the parameter h at the population level (average number of female mates per male) from our individual-level estimates of μ . If a population's average μ was ≤ 1 (i.e., on average, females were in scenarios 1 or 2), h was set to 1 because females were predominantly monogamous within-seasons (precisely what Reviewer 2 suggests should happen). However, when a population's average μ was >1 (i.e., on average females were in scenario 3), h was calculated as the inverse of the population average. This distinction is shown in the new Equation 6 of the revised manuscript (line 278).

In light of this, we have edited the following sections in our methods to clarify (lines 270-280):

*“Although both sexes can acquire multiple mates in a single breeding season, **within-season** polygamy is typically female biased in plovers. Thus, in accordance with the predominantly polyandrous or monogamous mating systems seen across these six populations, h was **derived** from the average annual number of mates per female (μ):*

$$\mu = \frac{1}{n} \sum_{i=1}^n \frac{m_i}{b_i} \quad (\text{Eq. 5})$$

*where, n is the total number of females in a given population, b is the total number of years female i was seen breeding, and m is the total number of mating partners female i had over b years. **Thus, if μ was less than or equal to one, females tended to have only one mating partner annually, and h was set to 1. Alternatively, if μ was greater than 1, females were polygamous and h was calculated as the inverse of μ :***

$$h = \begin{cases} 1, & \mu \leq 1 \\ \mu^{-1}, & \mu > 1 \end{cases} \quad (\text{Eq. 6})$$

Furthermore, we have modified the y-axis label and legend of Supplementary Figure 4 to clearly show μ (see below and on page 36 of the revised manuscript).

“Supplementary Figure 4. Variation in annual female mating rates (μ) among...” (line 602)

“...and black points are population averages ($\mu \pm 1$ SD).” (line 610)

2.3) Also, the perturbation analysis was a bit unclear to me. In line 321, the authors refer to “indirect effects of parameter perturbations on population growth.” Here I was wondering: do they really mean population growth, or do they mean ASR?

We thank Reviewer 2 for highlighting this. Perturbation analyses must incorporate the indirect effects of parameter perturbations on some measure of population “response” (i.e., growth, ASR, etc.) via their effects on population structure. In our study, ASR is the population response of interest, however, the approach we used could easily be employed for other population responses, such as the population growth rate (i.e., lambda), for example (as we did in an earlier, single population study: Eberhart-Phillips et al. 2017, PNAS 10.1073/pnas.1620043114). The key point is that parameter perturbations not only change baseline vital rates, but they also change the relative abundance of the sexes (‘population structure’) which can affect mating and regeneration; a rigorous perturbation analysis must account for both of these effects. To address this, we have replaced “growth” with “response” and elaborated on how perturbation analysis of a two-sex model indirectly affects population structure:

*“Perturbation analyses must therefore accommodate the indirect effects of parameter perturbations on population **response** via their effects on population structure, **such as the relative abundance of males and females which can affect mating dynamics and fecundity.**” (lines 337-340)*

2.4) Then they go on to say (line 322): “To estimate the sensitivities of the vital rate parameters to ASR...” Here I was wondering: do they really mean “sensitivities of the ASR with respect to vital rate parameters”? My understanding is that sensitivities are calculated for model outputs with respect to model inputs, rather than the other way round.

*Great point. We changed this to: “...the sensitivities **of the ASR to vital rate parameters**, we employed...” (line 340)*

2.5) Finally, the authors might profitably look at a recent paper (Fromhage & Jennions 2016; Nature Comm 7: 12517) that gives reasons for why the cause of ASR variation may actually matter. On the other hand, Kokko & Jennions’ 2008 prediction that “sex-specific survival creates unequal mating opportunities via ASR that may influence mating patterns and parenting strategies” was in part based on a technical flaw in their model. The corrected version by Fromhage & Jennions predicts no such effect for sex-specific survival during time-in, i.e., while searching for mates.

We thank Reviewer 2 for suggesting this article. It is indeed very relevant and we have therefore cited it in our discussion. Fromhage and Jennions (2016) present a compelling argument that sex-specific maturation rates could facilitate sex-biased mating opportunities and hence shape parental care patterns in ways that may differ than that expected under ASR variation. As our study did not discuss the role of time-in and time-out (i.e., operational sex ratio dynamics), we decided to cite Fromhage and Jennions (2016) in the context of the unexpected parental care patterns seen by C. pecuarius:

*“...or because of sex differences in parental quality (Amat et al. 2000) or the age at maturation (**Fromhage and Jennions, 2016**).” (lines 131-132)*

Regarding our citation of Kokko and Jennions (2008), we were careful to refer to this article in a context unrelated to the technical flaw in their model. On lines 152-154 we discuss the causality dilemma of ASR and parental care – a notion that was nicely presented by Kokko and Jennions (2008), and not directly linked to the error in their theoretical model (which

hinges on their statement on page 932 of their original article: “The future pay-off from desertion is directly proportional to how soon a parent can mate and leave the mating pool (i.e., the parental mating rate)”).

Reviewer #3 (Remarks to the Author):

3.1) This is a nice study using long-term data from six species of shorebird (one population per species).

*We thank Reviewer 3 for their positive comments of our study. However, as described in the “Results and Discussion” section (lines 80-82), our study contains six populations comprising five species (i.e., two populations were of the same species, *C. alexandrinus*).*

3.2) The authors show that bias in the adult sex ratio (ASR), when it occurs, is consistently caused by sex differences in juvenile mortality or emigration. There is some evidence that populations/species with a biased sex ratio have higher rates of uniparental care. Surprisingly, however, it is not generally the case that the most common sex is the caring parent. This latter point suggests that there is no (simple) causal relationship between ASR and parental care. The quadratic model is perhaps predicted a priori. But the conclusion that bias in ASR causes breakdown of cooperation between parents rely on the assumption that there is no alternative explanation for why *A. pecuarius* has obligate single-parental care. But does not an alternative seem much more likely given that the expected result based on ASR (biased towards females) is female-only care, and not male care as is apparently the case? In fact, the stronger a priori expectation that the common sex will be more likely to care is not supported by the data. This makes me doubt whether the logic holds up. ASR could still be important for parental cooperation, of course. But with five data points, and an unexpected direction with respect to which sex that care for offspring, the conclusion that ‘parental cooperation breaks down with unbalanced sex ratio’ (Abstract) does not really seem to be supported by data.

We agree with Reviewer 3 that there is no simple causal relationship between ASR and parental care. Other factors are very likely to contribute to the variation we observed in parental care regardless of ASR bias (e.g., predation pressure, breeding-habitat availability, sex-differences in energetic demands, etc.). We highlight these alternative explanations in the discussion of the manuscript on lines 128-132.

Reviewer 3 correctly points out that a more direct a priori prediction is that the scarcer sex is more likely to desert (a similar point raised in comment 1.12 by Reviewer 1). We also had this prediction in mind when conceiving our study, and state this in our manuscript (lines 110 – 113).

Considering Reviewer 3’s useful point, we have decided to conduct an additional analysis testing the relationship between ASR variation and the prevalence of male-only or female-only care (see below and Supplementary Figure 3 on page 35 of the revised manuscript).

The caption accompanying the new Supplementary Figure 3 is:

Supplementary Fig. 3. Relationship between uni-parental care and the adult sex ratio. (a) Predicted prevalence of male-only care (left panel) or female-only care (right panel) in response to adult sex ratio variation. (b) Observed relationship between parental care strategies and adult sex ratio estimates among the six studied populations. Faint white lines illustrate each iteration of the bootstrap, which randomly sampled an adult sex ratio and parental care estimate from each population’s uncertainty distribution and fitted them to a the a priori exponential model (Eq. 12). (c) Proportion of monitored plover families that exhibit parental cooperation (white) or uni-parental care by males (green) or females (orange). Sample sizes reflect the number of families monitored per population, circled numbers correspond to the data point labels shown in panel b. (lines 592-601)

Furthermore, we describe this additional analysis in our methods:

We also evaluated the relationship between ASR variation and male-only or female-only care using a similar bootstrap procedure of the following models:

$$P_{\sigma} = \beta_0 + \beta_1^A + \varepsilon \quad , \quad P_{\varphi} = \beta_0 - \beta_1^A + \varepsilon \quad (\text{Eq. 12})$$

where P_{σ} and P_{φ} are the proportions of families exhibiting male-only or female-only care, respectively. In this case, we chose exponential models a priori as we expected a non-linear increase in uni-parental care by the abundant sex under biased ASR (Supplementary Fig. 3a). This analysis demonstrated that male-only care tended to be more common in populations with male-biased ASR (mean $\beta_1 = 0.682$ [-0.366, 1.555 95% CI]) and female-only care tended to be more common in female-biased populations (mean $\beta_1 = -0.205$ [-0.502, 0.037 95% CI]; Supplementary Fig. 3b). However, the overall magnitude of the effect of ASR variation on female-only care was less than that of male-only care. (lines 364-374)

Curiously however, (rightfully noted by Reviewer 3) the unexpectedly high rate of female desertion in *C. pecuarius* (see, for example, the male-only panel in Supplementary Figure 3b) does not fit our prediction (although all 5 other populations do). We discuss this surprising result in the manuscript:

“Counterintuitively, we also found a high rate of male-only care in *C. pecuarius* despite ASR being female-biased (Fig. 3b). Yet, in line with expectations, *C. pecuarius* showed the highest proportion of female-only care among our studied populations (Fig. 3b, Supplementary Fig. 3b). This indicates partial support of a breeding strategy that is flexible to local mating opportunities provided by ASR bias, while also suggesting that other factors may play a role, such as the energetic costs of egg production **imposed on females** or because of sex differences in parental quality or the age at maturation” (lines 125-132)

In addition, please note that we also had the unusual opportunity to assess *intra*-specific variation in parental care under ASR variation (i.e., included in our study were two independent populations of *C. alexandrinus*. See comment 3.1 above). Comparison of these two populations supports the notion that ASR bias contributes towards a breakdown of parental cooperation. To highlight this within-species comparison, we have added the following text to our discussion:

“**Moreover, the relationship between parental cooperation and local ASR bias was apparent in our within-species contrast of *C. alexandrinus*: the unbiased Cape Verde population exhibited a higher rate of parental cooperation than the male-biased population in Turkey (Fig. 3a)**” (lines 122-125)

We are grateful that Reviewer 3 noted that the quadratic model was our a priori prediction, however, to make this more evident to the reader in our manuscript, we have edited the following texts:

“We evaluated **our a priori prediction of a quadratic relationship** between parental cooperation and **ASR variation using a regression analysis** incorporating a bootstrap procedure that acknowledged uncertainty in our estimates **of ASR and parental care** (see Methods for details).” (lines 115-118)

“To test the relationship between ASR bias and parental cooperation, we conducted a regression analysis of the following **quadratic** model:

$$P_{\sigma\varphi} = \beta_0 + \beta_1 A + \beta_2 A^2 + \varepsilon \quad (\text{Eq. 11})$$

where $P_{\sigma\varphi}$ is the proportion of families exhibiting parental cooperation, β_i are the regression parameters (i.e., intercept and coefficient), A is the ASR, and ε is random error. We chose a quadratic model a priori as we expected maximum parental cooperation at unbiased ASR but minimum cooperation at both male- and female-biased ASRs (see inset in Fig. 3a).” (lines 351-357)

To acknowledge the uncertainty associated with our limited sample size in this cross-species comparison, we have decided to calculate the multinomial confidence intervals of our point estimates of parental care (amended to Supplementary Table 2), and subsequently use these uncertainty distributions in the bootstrap regression procedure:

“To acknowledge uncertainty in these parental care proportions given variation in sample size, we used a method that estimated simultaneous 95% confidence intervals according to a multinomial distribution (Sison and Glaz 1995; Supplementary Table 2).” (lines 205-208)

*“This relationship was assessed with a bootstrap procedure that incorporated uncertainty in our estimates of ASR and parental care. Each iteration of the bootstrap (i) randomly sampled an ASR value from the 95% confidence interval of each population shown in Fig. 2b, (ii) **randomly sampled a parental care value from the truncated 95% confidence interval of each population shown in Supplementary Table 2, then (iii) fitted the regression model. We ran 1,000 iterations of the bootstrap and evaluated overall relationships by visualizing the central tendency of the regressions.**” (lines 358-364)*

3.3) Nevertheless, this is an elegant study (with beautiful figures!) and the hypothesis that the relationship between ASR and parental care is causal is interesting (but obviously itself not novel).

We thank Reviewer 3 and appreciate their kind comments.

Minor comments

3.4) Line 66. What do you mean by ‘life history origin(s) of ASR variation’? The stage at which mortality occurs?

This is a similar point made by Reviewer 1, please see our response to comment 1.2 above.

3.5) Line 89. I do not mind the term ‘apparent survival’ (and it does not seem crucial for the aims to that mortality is the cause of biased ASR), but the authors occasionally slip into using ‘survival’ (e.g., line 128) or ‘survivorship’ (e.g., line 547). It may be clumsier to read, but consistency is probably a good thing here.

We agree with Reviewer 3 that consistent terminology is important. We have made these edits throughout the manuscript when appropriate:

*“...sex differences in **apparent survival of juveniles.**” (lines 48-49)*

*“We found that sex biases in **apparent survival** during the juvenile stage contributed the most to sex ratio bias of the adult population: in populations with significantly skewed ASR, sex biases in juvenile **apparent survival** contributed on average 7.2 times more than sex biases in adult **apparent survival** and 24.6 times more than sex biases at hatching.” (lines 102-106)*

*“...where transition probabilities (ϕ) between life stages are the **apparent survival rates...**” (line 253)*

*“...sex differences in juvenile **apparent survival** made the largest overall contribution...” (lines 346-347)*

3.6) Line 135. What do you mean by ‘genotype-sex interactions’? Whatever it means, it seems entirely speculative (I could think of many reasons why males and females differ in survival or dispersal, none of which have to do with their genotype in any more interesting way than that females are ZW and males ZZ).

We acknowledge that this is speculation since the proximate causes of sex-specific mortality are completely unknown. However, we feel that our speculation is a plausible notion, as there are both experimental and correlational studies suggesting a role for (autosomal) genetic variation on sex-specific survival: Foerster et al. 2007 Nature 447:1107-1110, Kanfi et al. 2012 Nature 483:218-221, Pemberton et al. 1988 Evolution 42:921-934. We have added the Kanfi et al. (2012) citation to the manuscript since this study provided experimental evidence supporting this notion. Furthermore, we have modified this sentence to make it more evident that this concept is entirely speculative:

*“...implying that sex-differences in mortality are at least partly driven by intrinsic factors, **conceivably for example** via genotype-sex interactions (Küpper et al. 2010, Kanfi et al. 2012).” (lines 140-141)*

3.7) Methods. I was expecting to see an analyses of how much the ASR varies across years. It may be that the results are there, but if so they do not seem easy to extract from the text or tables. Annual variation in ASR seems like a very important means to also make more out of the relationship to parental care, although I do realise that the sample size is quite small.

We agree with Reviewer 3 that an analysis of annual variation in ASR and parental care would be a useful consideration. However, as they point out, our sample size restricted us from this finer-scale investigation. We pooled our brood observations across years and kept our sex-specific survival model parsimonious by assuming no annual variation in survival (albeit we did attempt to control for annual variation in detection probability (lines 218-219)).

*It is worth recognizing that for each population, our ASR estimates are the asymptotic values predicted by a model that assumes the population is at equilibrium. This is an important difference to an ASR estimate based on observed counts of males and females. Ideally, we could compare our ‘asymptotic ASR’ estimate to an ‘observed ASR’ estimate, however, as we have a subset of most populations individually marked, an estimate of ‘observed ASR’ based on our absolute numbers of males and females in our mark-recapture data is uninformative. Fortunately, the *C. nivosus* population provides a case whereby the vast majority of individuals are marked and annual (and seasonal) variation in ‘observed ASR’ estimates could be considered (please see Carmona-Isunza et al. 2017 Behav. Ecol. 28:523-532 for an assessment of temporal variation in ASR at the *C. nivosus* population). To address these points in the manuscript, we have added the following text and citation of the *C. nivosus* population:*

“Note that our method estimated ASR as the asymptotic value predicted under the assumption that each population was at equilibrium and thus we could not evaluate inter-annual variation in asymptotic ASR. Nonetheless, our model-derived ASR estimate of the *C. nivosus* population falls within annual count-based ASR estimates of this population (Carmona-Isunza et al. 2017), providing support that our method is robust. Count-based estimates of ASR from the remaining populations in our study are unfortunately uninformative due to our limited sample of marked individuals with known sex.” (lines 300-306)

Additional minor changes to text:

- Line 27: Deleted “, Western Bank” (incorrect coauthor address)
- Line 36: changed correspondent’s phone number: “-328” to “-424”
- Line 45: replaced “...data of...” with “...data on...”
- Line 54: changed “...life history...” to “...life-history...”
- Line 61: modified “...being frequent...” to “...being more frequent...”
- Line 68: modified “...sex-ratio bias...” to “...sex ratio bias...”
- Line 82: modified “We employed...” to “We then employed...”
- Line 84-85: modified “...the number of...” to “...the numbers of...”
- Line 93: modified “...sex bias in...” to “...sex biases in...”
- Line 94: modified “...within and between...” to “...within and among...”
- Line 108: modified “...Fisher’s notion...” to “...Fisher’s prediction...”
- Line 112: modified “...to be greatest due...” to “...to be greater than that of the scarcer sex due...”
- Line 124: modified “...exhibited higher rates...” to “...exhibited a higher rate...”
- Line 125-126: changed “...female desertion...” to “...male-only care...”
- Line 126: modified “...female-biased (Fig. 3b). Yet, in line with expectations, *C. pecuarius* showed...” to “...female-biased (Fig. 3b), although in line with expectations, *C. pecuarius* also showed...”
- Line 127: changed “...male desertion...” to “...female-only care...”

- Line 128: added citation "...Supplementary Fig. 3b)."
- Line 128-129: modified "This indicates partial support of a breeding strategy that is flexible to..." to "This provides partial support for the notion that breeding strategies may respond flexibly to..."
- Line 134-135: modified "...of birds..." to "...of many birds..."
- Line 136-138: modified "...presented here indicate partial support of this, whereby island populations of *C. alexandrinus* and the endemic *C. thoracicus* have reduced gene flow in contrast to mainland..." to "...presented here are partially consistent with this hypothesis, as island populations of *C. alexandrinus* and the endemic *C. thoracicus* have reduced gene flow relative to comparable mainland..."
- Line 147: modified "...grow slower..." to "...grow more slowly..."
- Line 153: modified "...selection which could..." to "...selection that could..."
- Line 153: modified "...males of females..." to "...males over females..."
- Line 159-160: modified "...original notion..." to "...original prediction..."
- Line 160: changed "...life history..." to "...life-history..."
- Line 161-162: modified "...consequences on parental..." to "...consequences for parental..."
- Line 181: changed "...alpha-numeric..." to "...alphanumeric..."
- Line 191: changed "...populations, although as..." to "...populations although, as..."
- Line 191-192: "...as broods get older, a variable percentage of male and female parents desert..." to "...as broods get older, male or female parents may desert..."
- Line 225: We realize that it is best to notate the most parameterized model, so we changed: "... (e.g., $p \sim \text{sex} * \text{year} + \text{stage}$)..." to "... (e.g., $p \sim \text{sex} * \text{year} * \text{stage}$)..."
- Line 232: changed "...sex specific..." to "...sex-specific..."
- Line 264: modified "... (see below)..." to "... (i.e. mating system, see below)..."
- Line 264-265: modified "...males respectively in..." to "...males, respectively, in..."
- Line 293: replaced "...each bootstrap..." with "...each iteration..."
- Line 300-301: modified "...1000 bootstraps..." to "...1,000 iterations..."
- Line 310: added "...and molecularly sexed..."
- Line 312: changed "...sex, year and age..." to "...sex, year, and age..."
- Line 336: modified "...depend..." to "...depends..."
- Line 376-378: changed "...R version *You Stupid Darkness*..." to "...R version *Kite-eating Tree*..."
- Line 503: removed "awarded"
- Line 504: modified "Fieldwork funded..." to "Fieldwork was funded..."
- Line 508-509: added grant to acknowledgments "... (GR3/10957), Biotechnology and Biological Sciences Research Council (BBS/B/05788) to ICC and TS."
- Line 509: changed "...Acadamy..." to "...Academy..."
- Line 509-511: added grant to acknowledgments "TB: Leverhulme Fellowship. TS: NKFIH-2558-1/2015 and the Wissenschaftskollegium zu Berlin."
- Line 514-515: changed "We are grateful for constructive comments kindly offered by..." to "Our manuscript benefitted from the constructive comments of..."
- Line 514-516: added "W. Forstmeier" and "three anonymous reviewers" to acknowledgements and removed "J. Laake"
- Line 515: removed redundant reference to the information presented on lines 375- 381 "Please see our GitHub repository (https://github.com/leberhartphillips/Plover_ASR_Matrix_Modeling) for a RMarkdown and all raw datasets needed to reproduce our modelling and analyses."
- Line 549: modified "...1000 iterations..." to "...1,000 iterations..."
- Line 552: modified "... = 0.43 [0.340,..." to "... = 0.430 [0.340,..."
- Line 553: changed "Relationship between the adult sex ratio and the tendency for parental cooperation." To "Relationship between parental cooperation and the adult sex ratio."
- Line 555: modified "...ratio estimate..." to "...ratio and parental care estimate..."
- Line 555-556: modified "...population's distribution (Fig. 2b) and fitted them to the point estimates of parental cooperation (white portions shown in panel b)." to "...population's uncertainty distribution and fitted them to the *a priori* quadratic model (shown in inset, Eq. 11)."
- Line 567: Modified "Sample size summary..." to "Summary of..."

Supplementary Table 2: removed the sample size columns for each parental care state because they are redundant. Added 95% confidence intervals to the point estimates of parental care. See comment 3.2 for more details.

Line 574: changed "...chicks..." to "...hatchlings..."

Line 600: modified "...reflect number..." to "...reflect the number..."

Line 611: changed "...mark-recapture modeling..." to "...mark-recapture modelling..."

Line 614: modified "...as labels on the x-axes." to "...as labels on the y-axes."

Revised figures and legends accompanying responses above:

Figure 1. Modelling the demographic pathway of adult sex ratio bias among plovers worldwide. (a) Location of the six study populations. *C. pecuarius*, *C. marginatus*, and *C. thoracicus* breed sympatrically in south-western Madagascar, whereas the two populations of *C. alexandrinus* are geographically disparate, inhabiting southern Turkey and the Cape Verde archipelago. The studied *C. nivosus* population is located on the Pacific coast of Mexico. All populations inhabit saltmarsh or seashore habitats characterized by open and flat substrates. (b) Schematic of the stage- and sex-specific demographic transitions of individuals from hatching until adulthood and their contributions to the adult sex ratio (depicted here is *C. nivosus*). The hatching sex ratio (ρ , proportion of male hatchlings) serves as a proxy for the primary sex ratio and allocates progeny to the male or female juvenile stage. During the juvenile ('juv') stage, a subset of this progeny will survive (ϕ) to recruit and remain as adults ('ad'). Dotted clusters illustrate how a cohort is shaped through these sex-specific demographic transitions to derive the adult sex ratio (mortality indicated by grey dots). The reproduction function, $R(n_{\sigma}, n_{\phi})$, is dependent on mating system and the frequency of available mates (see *Methods* for details).

Figure 3. Relationship between parental cooperation and the adult sex ratio. (a) Faint white lines illustrate each iteration of the bootstrap, which randomly sampled an adult sex ratio and parental care estimate from each population's uncertainty distribution and fitted them to the *a priori* quadratic model (shown in inset, Eq. 11). (b) Proportion of monitored plover families that exhibit parental cooperation (white) or single-parent care by males (green) or females (orange). Sample sizes reflect number of families monitored per population.

Supplementary Figure 1. Contributions of sex-specific parameters to adult sex ratio bias. These results are based on a life-table response experiment (LTRE) that compared the empirically-derived sex-specific model to hypothetical scenarios with no sex differences in demographic rates (top panel: female-only rates, bottom panel: male-only rates). ASR is the proportion of the adult population that is male, thus changes in female-biased parameters have a negative effect on ASR and consequently their LTRE statistics are negative. Notation: h = mating system index (Eq. 6), ρ = hatching sex ratio, Juvenile = sex-biased apparent survival of juveniles, Adult = sex-biased apparent survival of adults.

Supplementary Figure 3. Relationship between uni-parental care and the adult sex ratio. (a) Predicted prevalence of male-only care (left panel) or female-only care (right panel) in response to adult sex ratio variation. (b) Observed relationship between parental care strategies and adult sex ratio estimates among the six studied populations. Faint white lines illustrate each iteration of the bootstrap, which randomly sampled an adult sex ratio and parental care estimate from each population's uncertainty distribution and fitted them to the *a priori* exponential model (Eq. 12). (c) Proportion of monitored plover families that exhibit parental cooperation (white) or uni-parental care by males (green) or females (orange). Sample sizes reflect the number of families monitored per population, circled numbers correspond to the data point labels shown in panel b.

Supplementary Figure 4. Variation in annual female mating rates (μ) among the six plover populations.

Sample sizes indicate the number of individual females in each population that had at least two recorded breeding attempts with identified male(s) during the study. Values below one represent females that bred over multiple years with the same mate (i.e. between season monogamy), whereas values greater than one represent females that have had more than one mate per year (i.e. within season polyandry). Values equal to one represent individuals that have had one mate per year, but have switched mates between years (i.e. between season polyandry but within season monogamy). White data points illustrate individual females, and black points are population averages ($\mu \pm 1$ SD).

Reviewers' comments:

Reviewer #1 (Remarks to the Author):

The authors have satisfactorily addressed all my points and I have only a few minor issues.

I 133: The first sentence doesn't capture the content of the paragraph correctly. Instead, replacing 'potentially lead to a skewed ASR' with 'arise' would remove the discrepancy.

I 159--160: I respectfully disagree with one of the major conclusions of the manuscript: 'provide a comprehensive empirical test of Fisher's original prediction that influential sex biases arise in life-history stages beyond parental control'. Population-level unbiased sex-allocation does not exclude biased sex-allocation at the level of parents. Both can co-exist if biases average out at population level.

I 275--276: the indices i are missing at the variables b and m .

I 277--278: I do not understand why h was set to 1 in case of polygyny. Please explain more.

I 279: why not say 'females were polyandrous' instead of 'polygamous'? The former would be more consistent.

I 326: I might be mistaken but I think you also looked at the contribution of ρ and h , so replacing 'and' by 'or' and adding 'or monogamous mating system h '. The latter suggestion rests on my assumption that monogamy was the hypothetical model you used when calculating the contributions of h .

I 328: This sentence could profit from adding information of what the ASR is decomposed into, i.e. adding 'into the contributions....'

I 331: This section would profit from a bit more detail. For example, I am confused whether M_0 is the same as M but one focal parameter or whether all sex-differences have been removed.

Eq 8: I would appreciate an example with an actual parameter of the model.

I 335--345: please comment on whether fertility rates stabilized (rather than cycled or exhibited chaotic behaviour) over the course of 1000 time steps.

I 548: I suggest to add 'survival' between 'specific' and 'rates' to be consistent.

I 581 'thus changes in': I don't understand how this sentence fits in, because the upper panel plots the effect of survival rates compared to the female-only scenario while the lower panel those compared to the male-only scenario. So I do not see the link to 'female-biased parameters'.

fig 3b: I suggest to horizontally flip the upper right picture of a plover family such that it is aligned with the two pictures in the lower right part.

fig S1: Please add information on whether the column for h is derived by comparing the empirical based model to a setting of $h=1$. If the y-axis actually plots the sum of $C(\theta)$ over θ , please add reference to eq 8. Otherwise, please spell out the label.

Reviewer #2 (Remarks to the Author):

Although the authors have made a good overall effort to respond to the reviewers' comments, my view is that they did not resolve one important issue.

In my previous review I pointed out that, for the purpose of population projection, it is irrelevant whether birds keep the same partner for multiple years, or whether they find a new partner each year. The authors agreed with this in their response letter. I also pointed out that their calculation of h is affected by whether birds keep the same partner across multiple years. The authors denied this, noting that they imposed a maximum value of $h=1$ ('functional monogamy') whenever the average number of unique mates per year per female (μ) was smaller than or equal to 1. I agree that this procedure can sometimes produce the correct outcome. For example, a population in which $\mu=0.5$ for all females would correctly be treated as equivalent to a population in which $\mu=1$ for all females. However, the procedure does not always work correctly. For example, if some females with $\mu=2$ create additional broods, these broods must be accounted for regardless of whether there exist long-term monogamous females that push the population average below one. In other words, the present procedure has the flaw of letting long-term monogamy confound its estimate of within-season mating patterns. To avoid this problem, I suggest the authors impose a boundary condition on μ (to enforce $\mu \geq 1$ for each female), instead of imposing a boundary condition on h (to enforce $h \leq 1$).

Response to NCOMMS-17-19269-B Decision Notification

Title: "Demographic causes of adult sex ratio variation and their consequences for parental cooperation"

Tracking #: NCOMMS-17-19269-B

Authors: Eberhart-Phillips et al.

Note: our responses are shown in *italics* following each reviewer's comment, and edits are shown in **bold italics**. Revised versions of figures and their legends are found at the end of this document, starting on page 8.

Reviewer #1 (Remarks to the Author):

1.1) The authors have satisfactorily addressed all my points and I have only a few minor issues.

We appreciate the constructive comments from Reviewer 1 regarding our revised manuscript and referee response, thank you.

1.2) l 133: The first sentence doesn't capture the content of the paragraph correctly. Instead, replacing 'potentially lead to a skewed ASR' with 'arise' would remove the discrepancy.

We agree with Reviewer 1 and have changed the wording as suggested:

*"There are several ways in which sex-biased juvenile survival could **arise**." (line 135)*

1.3) l 159--160: I respectfully disagree with one of the major conclusions of the manuscript: 'provide a comprehensive empirical test of Fisher's original prediction that influential sex biases arise in life-history stages beyond parental control'. Population-level unbiased sex-allocation does not exclude biased sex-allocation at the level of parents. Both can co-exist if biases average out at population level.

This is a valid point. To acknowledge this, we have modified the following sections and have added a relevant citation pertinent to the issue of studying sex allocation at the individual or population level (i.e., Schindler et al. 2015, Nature 526:249):

*"Moreover, variation in hatching sex ratio had no effect on ASR **and remained unbiased even in populations with strong sex differences in juvenile survival. This provides empirical support for Fisher's¹³ prediction of unbiased sex allocation regardless of sex-biased survival of independent young or adults. However, we cannot dismiss biased sex allocation at the individual level, which would average out at the population level (Schindler et al. 2015). This critical test warrants further long-term study.**" (lines 106-111)*

AND

*"...and provide a comprehensive **population-level** test of Fisher's¹³ original prediction that influential sex biases arise in life-history stages beyond parental control." (lines 160-161)*

1.4) l 275--276: the indices i are missing at the variables b and m.

Thanks for pointing this out. We corrected this (lines 278-279).

1.5) l 277--278: I do not understand why h was set to 1 in case of polygyny. Please explain more.

*To clarify, we did not set h to 1 in the case of polygyny. Given that the members of our study system (*Charadrius spp.*) are predominantly polyandrous or monogamous, we chose to model mating dynamics from the female perspective. Thus, we measured mating system as the annual number of mates per female (i.e., μ , see Eq. 5). If μ was ≤ 1 , females had at most 1 mate per year, and therefore the most accurate mating system describing this is monogamy*

(i.e., $h = 1$). To clarify this in the manuscript, we have modified the following section (note this is a similar response to comment 2.1 from Reviewer 2):

“Although both sexes can acquire multiple mates in a single breeding season, within-season polygamy is typically female biased in plovers, **meaning that females tend to have multiple male partners within a season**. Thus, in accordance with the predominantly polyandrous or monogamous mating systems seen across these six populations, h was derived **by first calculating the mean** annual number of mates for each female (μ_i):

$$\mu_i = \begin{cases} 1, & \text{if } \frac{m_i}{b_i} \leq 1 \\ \frac{m_i}{b_i}, & \text{if } \frac{m_i}{b_i} > 1 \end{cases} \quad (\text{Eq. 4})$$

where, b_i is the total number of years female i was seen breeding and m_i is the total number of mating partners female i had over b_i years. If female i tended to have only one mating partner within- and between seasons (i.e., $\frac{m_i}{b_i} \leq 1$), μ_i was set to 1 because they were functionally monogamous. Alternatively, if female i was polyandrous within season, μ_i was greater than 1 to account for the additional fecundity of these extra matings. From this we calculated h as the inverse of the population average of μ_i :

$$h = \left(\frac{1}{n} \sum_{i=1}^n \mu_i \right)^{-1} \quad (\text{Eq. 5})$$

where, n is the total number of females in a given population.” (lines 272-285)

1.6) I 279: why not say 'females were polyandrous' instead of 'polygamous'? The former would be more consistent.

Thanks for pointing this out. We agree that the suggested wording would be more informative and have made this modification:

“Alternatively, if female i was **polyandrous** within seasons...” (lines 281)

1.8) I 326: I might be mistaken but I think you also looked at the contribution of ρ and h , so replacing 'and' by 'or' and adding 'or monogamous mating system h '. The latter suggestion rests on my assumption that monogamy was the hypothetical model you used when calculating the contributions of h .

Thanks for noting this. We indeed considered the contributions of hatching sex ratio (ρ) and mating system (h). To clarify, we have changed the text to read:

“...were set equal to the male rates (or vice versa), the hatching sex ratio was unbiased (i.e. $\rho = 0.5$), **and mating system was monogamous (i.e. $h = 1$)**.” (lines 330-332)

1.9) I 328: This sentence could profit from adding information of what the ASR is decomposed into, i.e. adding 'into the contributions....'

We agree that clarification may help understanding here. We have modified the text to read:

“...our LTRE identifies the drivers of ASR bias by decomposing **the absolute parameter contributions to the difference**...” (lines 332-333)

1.10) I 331: This section would profit from a bit more detail. For example, I am confused whether M_0 is the same as M but one focal parameter or whether all sex-differences have been removed.

M_0 is a matrix in which all sex-differences have been removed. To clarify, we have reworded this as:

“Our LTRE compared the observed scenario (M), to a **null scenario with no sex differences** (M_0), whereby all **male** survival rates were set equal to the **female** rates

$(M_{0\varphi})$, the hatching sex ratio was unbiased (i.e. $\rho = 0.5$), and mating system was monogamous (i.e. $h = 1$)." (lines 329-332)

1.11) Eq 8: I would appreciate an example with an actual parameter of the model.

As requested, we provide Reviewer 1 an example of how the contributions were calculated (i.e., following Eq. 8). We have decided not to amend this to the article because examples of LTRE are shown elsewhere in the literature (e.g., Chapter 10 of Caswell 2001, 2nd edition; Veran and Beissinger, 2009 Ecology Letters 12:129). However, if Reviewer 1 or the editor feels that providing this example to readers may improve the clarity of our article, we would be happy to include it as Supplementary Material.

In this example, we look at juvenile apparent survival in the C. nivosus population under the $M_{0\sigma}$ scenario:

1) Here is the M matrix:

$$\mathbf{M} = \begin{bmatrix} 0 & R_{\varphi} \times (1 - 0.469) & 0 & R_{\sigma} \times (1 - 0.469) \\ 0.088 & 0.684 & 0 & 0 \\ 0 & R_{\varphi} \times 0.469 & 0 & R_{\sigma} \times 0.469 \\ 0 & 0 & 0.147 & 0.696 \end{bmatrix}$$

2) Thus, the $M_{0\sigma}$ matrix is (all survival rates = male rates, $k = 3$, $h = 1$, $\rho = 0.5$; note that fecundities are derived numerically using the mating functions shown in Eq. 4):

$$\mathbf{M}_0 = \begin{bmatrix} 0 & R_{\varphi} \times (1 - 0.5) & 0 & R_{\sigma} \times (1 - 0.5) \\ 0.147 & 0.696 & 0 & 0 \\ 0 & R_{\varphi} \times 0.5 & 0 & R_{\sigma} \times 0.5 \\ 0 & 0 & 0.147 & 0.696 \end{bmatrix}$$

3) Calculate the M' matrix (Eq 9) as the "halfway" matrix between M and M_0 :

$$\mathbf{M}' = \begin{bmatrix} 0 & R_{\varphi} \times (1 - 0.485) & 0 & R_{\sigma} \times (1 - 0.485) \\ 0.117 & 0.689 & 0 & 0 \\ 0 & R_{\varphi} \times 0.485 & 0 & R_{\sigma} \times 0.485 \\ 0 & 0 & 0.147 & 0.696 \end{bmatrix}$$

4) Estimate the sensitivity of ASR to female juvenile survival in M' (i.e., cell [2, 1] in the matrix above). First, we perturbed the survival rate from 0 to 1 (grey points in plot below) while keeping all other vital rates in M' constant, and derived ASR numerically through simulation over 1000 time-steps. Second, we estimated the $f(x)$ of this trend (black line in plot below). Third, we estimated the tangent of $f(x)$ at the actual value of female juvenile survival (blue dot; 0.147). The slope of the tangent is the sensitivity, in this case -2.125.

5) Next we simply calculate the difference between female and male juvenile survival in M (i.e., the first part of Eq. 8):

$$0.088 \quad 0.147 \quad 0.059$$

6) Finally, we can estimate the contribution that juvenile survival has to ASR bias under the "Male M_0 scenario" as the difference in step 5) multiplied by the sensitivity in step 4):

0 059 2 125 0 125

7) This value (0.125) is the light grey value shown in the bottom left-most panel of Figure S1:

To further clarify Eq. 8, we have provided additional information and formulae pertinent to the $M_{0\sigma}$ and $M_{0\varphi}$ scenarios:

“Our LTRE compared the observed scenario (M), to a null scenario with no sex differences (M_0), whereby all male survival rates were set equal to the female rates ($M_{0\varphi}$), the hatching sex ratio was unbiased (i.e. $\rho = 0.5$), and mating system was monogamous (i.e. $h = 1$). Thus, our LTRE identifies the drivers of ASR bias by decomposing the absolute parameter contributions to the difference between the ASR predicted by our model and an unbiased ASR¹⁸. To verify if our method was robust, we also evaluated a null scenario in which all female survival rates were set equal to the male rates ($M_{0\sigma}$).

The LTRE contributions (C) of a sex bias in stage-specific apparent survival (ϕ_i), mating system (h), and hatching sex ratio (ρ) in M were calculated following Veran and Beissinger¹⁸:

$$\begin{aligned}
 M_{0\sigma} \text{ scenario: } C(\phi_i) &= (\phi_{\varphi i} - \phi_{\sigma i}) \times \left. \frac{\partial ASR}{\partial \phi_{\varphi i}} \right|_{M'} \\
 M_{0\varphi} \text{ scenario: } C(\phi_i) &= (\phi_{\sigma i} - \phi_{\varphi i}) \times \left. \frac{\partial ASR}{\partial \phi_{\sigma i}} \right|_{M'} \\
 M_{0\sigma} \text{ and } M_{0\varphi} \text{ scenarios: } C(h) &= (h - 1) \times \left. \frac{\partial ASR}{\partial h} \right|_{M'} \\
 M_{0\sigma} \text{ and } M_{0\varphi} \text{ scenarios: } C(\rho) &= (\rho - 0.5) \times \left. \frac{\partial ASR}{\partial \rho} \right|_{M'} \quad (\text{Eq. 8})
 \end{aligned}$$

where $\left. \frac{\partial ASR}{\partial \theta} \right|_{M'}$ is the sensitivity of ASR to perturbations in the demographic rate θ (i.e. ϕ_i , h , or ρ) of matrix M' , which is a reference matrix “midway” between the observed and the null scenarios...” (lines 329-345)

Under either scenario the sum of all contributions is approximately equal to the difference between the ASR of M and 0.5 (i.e., the ASR of M_0). This follows the logic presented in formula 10.2 on page 261 of Caswell 2001, 2nd edition, whereby:

$$M_{0\sigma} \text{ scenario: } ASR_M \approx ASR_{M_{0\sigma}} + C(\phi_J)_{M_{0\sigma}} + C(\phi_A)_{M_{0\sigma}} + C(h) + C(\rho)$$

$$M_{0\varphi} \text{ scenario: } ASR_M \approx ASR_{M_{0\varphi}} + C(\phi_J)_{M_{0\varphi}} + C(\phi_A)_{M_{0\varphi}} + C(h) + C(\rho)$$

Furthermore, we have updated the y-axis title of Figure S1 to be more specific about the male and female scenarios of the LTRE (see end of document for revised figure).

1.12 | 335--345: please comment on whether fertility rates stabilized (rather than cycled or exhibited chaotic behaviour) over the course of 1000 time steps.

Fertility rates stabilized for all simulations of all populations. Here is an example of the sex-specific fertility rates over the first 100 time steps from a random bootstrap of each of the six populations (note: each iteration ran for 1000 time steps, however fertility stabilized within the first 100 time steps, which are shown here):

1.13) I 548: I suggest to add 'survival' between 'specific' and 'rates' to be consistent.

Thanks for this suggestion. We have edited this as:

“...the sex- and stage-specific **apparent survival** rates shown...” (line 563)

1.14) I 581 'thus changes in ...': I don't understand how this sentence fits in, because the upper panel plots the effect of survival rates compared to the female-only scenario while the lower panel those compared to the male-only scenario. So I do not see the link to 'female-biased parameters'.

We agree with Reviewer 1 that our original wording was confusing. Our intent was to explain what a negative contribution means. Because our measure of ASR is defined as the proportion of the adult population that is male, demographic rates that are female-biased will have a negative contribution. To clarify this, we have edited this caption as:

“...model to **null** scenarios with no sex differences in demographic rates (top panel: **M_0 consists of female rates**, bottom panel: **M_0 consists of male rates; Eq. 8) and a monogamous mating system (i.e. $h = 1$). Because ASR is measured as the proportion of the adult population that is male, **LTRE statistics are negative for demographic rates that are female-biased in each population.**” (lines 595-599)**

1.15) fig 3b: I suggest to horizontally flip the upper right picture of a plover family such that it is aligned with the two pictures in the lower right part.

We agree with Reviewer 1 that it would be best to keep the alignment of these silhouettes consistent and have modified the figure as suggested. Please see this correction in the revised Fig 3b at the end of this document.

1.16) fig S1: Please add information on whether the column for h is derived by comparing the empirical based model to a setting of $h=1$. If the y-axis actually plots the sum of $C(\text{teta})$ over teta, please add reference to eq 8. Otherwise, please spell out the label.

We have modified the caption of Fig S1 to include information about h:

“...in demographic rates (top panel: M_0 consists of female rates, bottom panel: M_0 consists of male rates; Eq. 8) and a monogamous mating system (i.e., $h = 1$).”
(lines 595-597)

Furthermore, we have modified the y-axis title to specify that these are the absolute contributions to adult sex ratio bias: “**Absolute contribution to adult sex ratio bias**”. Please see the revised plot at the end of this document and our response to comment 1.11) above for more details.

Reviewer #2 (Remarks to the Author):

2.1) Although the authors have made a good overall effort to respond to the reviewers’ comments, my view is that they did not resolve one important issue. In my previous review I pointed out that, for the purpose of population projection, it is irrelevant whether birds keep the same partner for multiple years, or whether they find a new partner each year. The authors agreed with this in their response letter. I also pointed out that their calculation of h is affected by whether birds keep the same partner across multiple years. The authors denied this, noting that they imposed a maximum value of $h=1$ (“functional monogamy”) whenever the average number of unique mates per year per female (μ) was smaller than or equal to 1. I agree that this procedure can sometimes produce the correct outcome. For example, a population in which $\mu=0.5$ for all females would correctly be treated as equivalent to a population in which $\mu=1$ for all females. However, the procedure does not always work correctly. For example, if some females with $\mu=2$ create additional broods, these broods must be accounted for regardless of whether there exist long-term monogamous females that push the population average below one. In other words, the present procedure has the flaw of letting long-term monogamy confound its estimate of within-season mating patterns. To avoid this problem, I suggest the authors impose a boundary condition on μ (to enforce $\mu \geq 1$ for each female), instead of imposing a boundary condition on h (to enforce $h \leq 1$).

We appreciate the constructive criticism of our mating function by Reviewer 2. Our original approach worked well for plovers because of their complex within vs. between season mating system, however we acknowledge that this method is less general can sometimes be misleading. Therefore, we agree with Reviewer 2’s suggestion to “impose a boundary condition on μ (to enforce $\mu \geq 1$ for each female)”.

When we did as Reviewer 2 advised and forced females with long-term monogamy to have $\mu = 1$, the population averages of h became more polyandrous, thus acknowledging the additional fecundity created by the polyandrous subset of a given population. Nonetheless, all subsequent results effectively stayed the same after this modification – which was expected given the extremely low sensitivity of ASR to h in our matrix model (i.e., Supplementary Fig. 1). Note however that our uncertainty in ASR slightly increased (i.e., revised 95% CIs of ASR in Fig. 2b and Fig. 3a at the end of this document). To clarify this modification of the mating function, we have re-written this section as follows:

*“Although both sexes can acquire multiple mates in a single breeding season, within-season polygamy is typically female biased in plovers, **meaning that females tend to have multiple male partners within a season**. Thus, in accordance with the predominantly polyandrous or monogamous mating systems seen across these six populations, h was derived **by first calculating the mean** annual number of mates for each female (μ_i):*

$$\mu_i = \begin{cases} 1, & \text{if } \frac{m_i}{b_i} \leq 1 \\ \frac{m_i}{b_i}, & \text{if } \frac{m_i}{b_i} > 1 \end{cases} \quad (\text{Eq. 4})$$

where, b_i is the total number of years female i was seen breeding and m_i is the total number of mating partners female i had over b_i years. If female i tended to have only one mating partner within- and between seasons (i.e., $\frac{m_i}{b_i} \leq 1$), μ_i was set to 1 because they were functionally monogamous. Alternatively, if female i was polyandrous within season, μ_i was greater than 1 to account for the additional fecundity of these extra matings. From this we calculated h as the inverse of the population average of μ_i :

$$h = \left(\frac{1}{n} \sum_{i=1}^n \mu_i \right)^{-1} \quad (\text{Eq. 5})$$

where, n is the total number of females in a given population.

Our dataset to estimate h for each population only included females for which we were confident of the identity of their mates, and had observed them in at least two reproductive attempts. In summary, h varied among populations (Supplementary Fig. 4), with *C. pecuarius* ($h = 0.80$), *C. nivosus* ($h = 0.80$), and *C. alexandrinus* (Turkey; $h = 0.81$) having **more** polyandrous mating systems and *C. marginatus* ($h = 0.90$), *C. alexandrinus* (Cape Verde; $h = 0.96$), and *C. thoracicus* ($h = 1$) all having more monogamous mating systems.” (lines 272-291)

Furthermore, we updated Fig S4 (see end of document) and its caption to reflect the changes to μ suggested by Reviewer 2:

“Values below one represent females that bred over multiple years with the same mate (i.e. **long-term** monogamy), whereas values greater than one represent females that have had more than one mate per year (i.e. within season polyandry). Values equal to one represent individuals that have had one mate per year, but have switched mates between years (i.e. between season polyandry but within season monogamy). White data points illustrate individual **females’ mates per year (i.e., m_i/b_i in Eq. 4)**, and black points are population averages **corrected for long-term monogamy according to Eq. 4** ($\mu \pm 1$ SD).” (lines 622-628)

Additional minor changes to text:

Line 96: replaced “...rendered significant deviations...” with “...rendered notable deviations...”

Line 97: removed hyphens in “male biased” and “female biased”

Line 104: removed “in populations with significantly skewed ASR,”

Line 105: updated values based on the modifications addressed in response 1.11: “sex biases in juvenile survival contributed on average 7.8 times more than sex biases in adult apparent survival and 326.6 times more than sex biases at hatching (Supplementary Fig. 1).”

Line 128: removed hyphen in “female biased”

Line 141-142: removed hyphen in “sex differences”

Lines 181 and 183: added citation to “Supplementary Video 1” to provide graphical insights of our field methods

Line 257: modified “...either male (i.e. ρ) or female (i.e. $1 - \rho$)...” to “...either male (ρ) or female ($1 - \rho$)...”

Line 260: modified “...time step of the model given...” to “...time step given...”

Line 265-266: modified “...mates acquired per female (i.e. mating system...)” to “...mates acquired per male (i.e. mating system...)”

Line 268: modified “Quantifying mating system” to “Quantifying the mating system”

Line 280: removed hyphen in “...within and between...”

Line 348: replaced “fertility” with “fecundity” to keep terminology consistent throughout manuscript.

Line 355: updated notation of $\frac{\partial \text{ASR}}{\partial \theta}$ to $\frac{\partial \text{ASR}}{\partial \theta} \Big|_{\mathbf{M}_r}$

Line 358: added: “Under either scenario (i.e. $\mathbf{M}_{0\text{♀}}$ or $\mathbf{M}_{0\text{♂}}$), our ...”

Lines 383-387: updated regression statistics following the modifications addressed in response 2.1: “This analysis demonstrated that male-only care tended to be more common in populations with male-biased ASR (mean $\beta_1 = 0.551$ [-0.849, 1.559 95% CI]) and female-only care tended to be more common in female-biased populations (mean $\beta_1 = -0.149$ [-0.427, 0.082 95% CI]; Supplementary Fig. 3b).”

Line 529: added funder to acknowledgments: “...ÉLVONAL-KKP 126949...”

Line 633: changed “Supplementary Movie 1” to “Supplementary Video 1”

Revised figures and legends accompanying responses above:

Figure 2. Inter- and intra-specific variation in sex-biased demography. (a) Hatching sex ratios of successful clutches (proportion of chicks that are male) are shown as point estimates ($\rho \pm 95\%$ CI; left y-axis), and sex bias (i.e. difference between males and females) in annual apparent survival rates of juveniles (ϕ_{juv}) and adults (ϕ_{ad}) are shown as violin plots (right y-axis). Horizontal lines within violin plots indicate the median and interquartile ranges of the bootstrapped estimates (see *Methods* for details). (b) Bootstrap distributions of the derived ASRs based on the sex- and stage-specific apparent survival rates shown in panel a. Vertical bars on the right side of histograms indicate the 95% CI of ASRs based on 1,000 iterations of the bootstrap (mean ASR [95% CI]): *C. nivosus* = 0.638 [0.464, 0.788], *C. alexandrinus* [Turkey] = 0.576 [0.487, 0.659], *C. alexandrinus* [Cape Verde] = 0.463 [0.339, 0.587], *C. thoracicus* = 0.401 [0.086, 0.716], *C. marginatus* = 0.434 [0.328, 0.546], *C. pecuarius* = 0.363 [0.220, 0.512].

Figure 3. Relationship between parental cooperation and the adult sex ratio. (a) Faint white lines illustrate each iteration of the bootstrap, which randomly sampled an adult sex ratio and parental care estimate from each population's uncertainty distribution and fitted them to the *a priori* quadratic model (shown in inset, Eq. 10). (b) Proportion of monitored plover families that exhibit parental cooperation (white) or single-parent care by males (green) or females (orange). Sample sizes reflect number of families monitored per population.

Supplementary Figure 1. Contributions of sex-specific parameters to adult sex ratio bias. These results are based on a life-table response experiment (LTRE) that compared the empirically-derived sex-specific model to null scenarios with no sex differences in demographic rates (top panel: \mathbf{M}_0 consists of female rates, bottom panel: \mathbf{M}_0 consists of male rates; Eq. 8) and a monogamous mating system (i.e., $h = 1$). Because ASR is measured as the proportion of the adult population that is male, LTRE statistics are negative for demographic rates that are female-biased in a given population. Notation: h = mating system index (Eq. 6), p = hatching sex ratio, Juvenile = sex-biased apparent survival of juveniles, Adult = sex-biased apparent survival of adults.

Supplementary Figure 3. Relationship between uni-parental care and the adult sex ratio. (a) Predicted prevalence of male-only care (left panel) or female-only care (right panel) in response to adult sex ratio variation. (b) Observed relationship between parental care strategies and adult sex ratio estimates among the six studied populations. Faint white lines illustrate each iteration of the bootstrap, which randomly sampled an adult sex ratio and parental care estimate from each population's uncertainty distribution and fitted them to the *a priori* exponential model (Eq. 12). (c) Proportion of monitored plover families that exhibit parental cooperation (white) or uni-parental care by males (green) or females (orange). Sample sizes reflect the number of families monitored per population, circled numbers correspond to the data point labels shown in panel b.

Supplementary Figure 4. Variation in annual female mating rates (μ) among the six plover populations.

Sample sizes indicate the number of individual females in each population that had at least two recorded breeding attempts with identified male(s) during the study. Values below one represent females that bred over multiple years with the same mate (i.e. long-term monogamy), whereas values greater than one represent females that have had more than one mate per year (i.e. within season polyandry). Values equal to one represent individuals that have had one mate per year, but have switched mates between years (i.e. between season polyandry but within season monogamy). White data points illustrate individual females' mates per year (i.e., m_i/b_i in Eq. 4), and black points are population averages corrected for long-term monogamy according to Eq. 4 ($\mu \pm 1$ SD).